# Simulation of reactive solute transport in the critical zone: A Lagrangian model for transient flow and preferential transport

Alexander Sternagel[1], Ralf Loritz[1], Julian Klaus[2], Brian Berkowitz[3], Erwin Zehe[1]

[1] Karlsruhe Institute of Technology (KIT), Institute of Water Resources and River Basin Management, Hydrology, Germany
[2] Luxembourg Institute of Science and Technology (LIST), Environmental Research and Innovation Department, Catchment and Eco-Hydrology Research Group, Luxembourg
[3] Department of Earth and Planetary Sciences, Weizmann Institute of Science, Israel

*Correspondence to*: Alexander Sternagel (alexander.sternagel@kit.edu)

**Abstract.** We present a method to simulate fluid flow with reactive solute transport in structured, partially saturated soils using a Lagrangian perspective. In this context, we extend the scope of the Lagrangian Soil Water and Solute Transport Model (LAST) (Sternagel et al., 2019) by implementing vertically variable, non-linear sorption and first-order degradation processes during transport of reactive substances through a partially saturated soil matrix and macropores. For sorption, we develop an explicit mass transfer approach based on Freundlich isotherms because the common method of using a retardation factor is not applicable in the particle-based approach of LAST. The reactive transport method is tested against data of plot- and field-scale irrigation experiments with the herbicides Isoproturon and Flufenacet at different flow conditions over various periods. Simulations with HYDRUS 1-D serve as an additional benchmark. At the plot-scale, both models show equal performance at a matrix flow dominated site, but LAST better matches indicators of preferential flow at a macropore flow dominated site. Furthermore, LAST successfully simulates the effects of adsorption and degradation on the breakthrough-behaviour of Flufenacet with preferential leaching and remobilization. The results demonstrate the feasibility of the method to simulate reactive solute transport in a Lagrangian framework, and highlight the advantage of the particle-based approach and the structural macropore domain to simulate solute transport as well as to cope with preferential bypassing of topsoil and subsequent re-infiltration into the subsoil matrix.

## 1 Introduction

Reactive substances like pesticides are subject to chemical reactions within the critical zone (Kutílek and Nielsen, 1994; Fomsgaard, 1995). Their mobility and life span depend greatly on various factors like (i) the spectrum of transport velocities, (ii) the sorption to soil materials (Knabner et al., 1996) and (iii) microbial degradation and turnover (cf. Sect. 3). The multitude and complexity of these factors are a considerable source of uncertainty in pesticide fate modelling. It is still not fully understood how pesticides are transported within different soils and particularly how preferential flow through macropores impacts the breakthrough of these substances into streams and groundwater (e.g. Flury, 1996; Arias-Estévez et al., 2008; Frey et al., 2009; Klaus et al., 2014).

To advance our understanding of reactive solute transport (RT) of pesticides, particularly the joint controls of macropores, sorption and degradation, a combination of predictive models and plot-scale experiments is often used (e.g. Zehe et al., 2001; Simunek et al., 2008; Radcliffe and Simunek, 2010; Klaus and Zehe, 2011; Klaus et al., 2013). Such methods allow the assessment of the environmental risks arising from the wide use of reactive substances (Pimentel et al., 1992; Carter, 2000; Gill and Garg, 2014; Liess et al., 1999). Combining the Richards

and advection-dispersion equations is one common approach used to simulate water flow dynamics and (reactive) solute transport in the partially saturated soil zone. This approach has been implemented, for example, in the well-established models HYDRUS (Gerke and van Genuchten, 1993; Simunek et al., 2008), MACRO (Jarvis and Larsbo, 2012) and Zin AgriTra (Gassmann et al., 2013). However, this approach has well-known deficiencies in

simulating preferential macropore flow and imperfect mixing with the matrix in the vadose zone (Beven and Germann, 2013). As both processes essentially control environmental risk due to transport of reactive substances, a range of adaptions has been proposed to improve this deficiency (Šimůnek et al., 2003). One frequently-used adaption is the dual-domain concept, which describes matrix and macropore flow in separated, exchanging continua to account for local disequilibrium conditions (Gerke, 2006). However, studies show that even these dual-

domain models can be insufficient to quantify preferential solute breakthrough into the subsoil (Sternagel et al., 2019) or into tile drains (Haws et al., 2005; Köhne et al., 2009b, a). A different approach is to represent macropores as spatially connected, highly permeable flow paths in the same domain as the soil matrix (Sander and Gerke, 2009). This concept has been shown to operate well for preferential flow of water and bromide tracers at a forested hillslope (Wienhöfer and Zehe, 2014), and for bromide and Isoproturon transport through worm burrows into a

tile drain at a field site (Klaus and Zehe, 2011). Nevertheless, this approach is based on the Richards equation and is thus limited to laminar flow conditions with sufficiently small flow velocities corresponding to a Reynolds number smaller 10 (e.g. Bear, 2013; Loritz et al., 2017).

Particle-based approaches offer a promising alternative to simulate reactive transport. These approaches work with a Lagrangian perspective on the movement of solute particles in a flow field, rather than solving the advection-

dispersion equation directly. They have been particularly effective in quantifying solute transport alone, while the movement of the fluid carrying solutes is still usually integrated in systems based on Eulerian control volumes (e.g. Delay and Bodin, 2001; Zehe et al., 2001; Berkowitz et al., 2006; Koutsoyiannis, 2010; Klaus and Zehe, 2010; Wienhöfer and Zehe, 2014). In the context of saturated flow in fractured and heterogeneous aquifers, Lagrangian descriptions of fluid flow are already commonly and successfully applied. For example, the

Continuous Time Random Walk (CTRW) approach accounts for non-Fickian transport of tracer particles within the water flow through heterogeneous, geological formations via different flow paths with an associated distribution of velocities and thus travel times (Berkowitz et al., 2006; Berkowitz et al., 2016; Hansen and Berkowitz, 2020). However, Lagrangian modelling of fluid flow in the vadose zone is more challenging due to the dependence of the velocity field on the temporally changing soil moisture states and boundary conditions. This

explains why only a relatively small number of models use Lagrangian approaches for solute transport, and also for water particles (also called water "parcels") to characterize the fluid phase itself (e.g. Ewen, 1996a, b; Bücker-Gittel et al., 2003; Davies and Beven, 2012; Zehe and Jackisch, 2016; Jackisch and Zehe, 2018). Sternagel et al. (2019) proposed that these water particles may optionally carry variable solute masses to simulate non-reactive transport. Their Lagrangian Soil Water and Solute Transport Model (LAST) combines the assets of the Lagrangian

approach with an Euler-Grid to simulate fluid motion and solute transport in heterogeneous, partially saturated 1-D soil domains. It allows discrete water particles to travel at different velocities and carry temporally-variable solute masses through the subsurface domain. The soil domain is subdivided into a soil matrix and a structurally defined preferential flow/macropore domain (cf. Sect. 2). A comparison of HYDRUS 1-D and the LAST-Model based on plot scale tracer experiments showed that both models perform similarly in case of matrix flow dominated

tracer transport; however, under preferential flow conditions, LAST better matched observed tracer profiles indicating preferential flow (Sternagel et al., 2019).

While the results of Sternagel et al. (2019) demonstrate the feasibility of the Lagrangian approach to simulate conservative tracer transport even under preferential flow conditions during one-day simulations, a generalization of the Lagrangian approach to reactive solute transport and larger time scales is still missing.

The main objectives of this study are thus to

1. develop a method for reactive transport, i.e., sorption and degradation of solutes within the Lagrangian framework under well-mixed and preferential flow conditions and implement this into the LAST-Model. We initially test the feasibility of the method by simulating plot-scale experiments with a bromide tracer and the herbicide Isoproturon (IPU) during 2 days (Zehe and Flühler, 2001), and use corresponding simulations of the commonly applied model HYDRUS 1-D as a benchmark;

2. perform plot-scale simulations to explore the transport behaviour of bromide and IPU with the Lagrangian approach over 7 and 21 days to evaluate its performance on longer time scales. For this purpose, we make use of data from another plot-scale irrigation experiment (Klaus et al., 2014);

3. conduct simulations of breakthrough experiments with Flufenacet (FLU) on a tile-drain field site over a period of three weeks (Klaus et al., 2014), to examine the breakthrough behaviour and remobilization of reactive substances.

## 2 The LAST-Model: Concept, theoretical background and numerical implementation

### 2.1 Model concept

The LAST-Model combines a Lagrangian approach with an Euler-Grid to simulate fluid motion and solute transport in heterogeneous, partially saturated 1-D soil domains. Discrete water particles with a constant water mass and volume carry temporally variable information about their position and solute concentrations through defined domains for soil matrix and macropores that are subdivided into vertical grid elements (Euler-Grid). Prior to simulation, the initial water content of each grid element is converted to a corresponding water mass with the grid element volume and water density. The water mass of each grid element is summed to a total water mass in the entire soil domain and then divided by the total number of particles. In this way, the water particles in the soil domain are initially defined by a certain water mass. During the simulation, the number of water particles is counted in each time step and a new particle density per grid element is computed. By multiplying this water particle density with the particle mass and water density, a new soil water content per grid element and time step can be obtained (Zehe and Jackisch, 2016). Different fractions of the water particles in a grid element correspond to the subscale distribution of the water content among soil pores of different sizes. Consequently, different water particle fractions travel at different velocities (cf. Fig. 1). Their displacements are determined by the hydraulic conductivity and water diffusivity in combination with a spatial random walk (cf. Sect. 2.2, Eq. 5). This approach accounts for the joint effects of gravity and capillary forces on water flow in partially saturated soils. The use of an Euler-Grid allows for the necessary updating of soil water contents based on changing particle densities and related time-dependent changes in the velocity field. The space domain approach also reflects the fact that spatial concentration patterns and thus travel distances are usually observed in the partially saturated zone. The Euler-

Grid is hence necessary to calculate spatial concentration profiles and to properly describe specific interactions between matrix and macropore domain.

## 2.2 Underlying theory and model equations

### 2.2.1 Transient fluid flow in the partially saturated zone

The LAST-Model (Sternagel et al., 2019) is based on the Lagrangian approach of Zehe and Jackisch (2016), which was introduced to simulate infiltration and soil water dynamics in the partially saturated zone using a non-linear random walk in the space domain. The results of test simulations confirmed the ability of the Lagrangian approach to simulate water dynamics under well-mixed conditions in different soils, in good accord with simulations using a Richards equation solver. We refer the reader to the study of Zehe and Jackisch (2016) for further details on the
model concept.

*Derivation of particle displacement equation*

Our starting point is the soil-moisture-based form of the Richards equation:

$$\frac{\partial \theta}{\partial t} = \frac{\partial K(\theta)}{\partial z} + \frac{\partial}{\partial z}\left(D(\theta)\frac{\partial \theta}{\partial z}\right),\tag{1}$$

$$\text{with } D(\theta) = K(\theta)\,\frac{\partial \Psi}{\partial \theta}.$$

By multiplying the hydraulic conductivity $K$ in the first term of Eq. 1 by $\frac{\theta}{\theta}$ (= 1), we obtain:

$$\frac{\partial \theta}{\partial t} = \frac{\partial}{\partial z}\left[\frac{K(\theta)}{\theta}\,\theta\right] + \frac{\partial}{\partial z}\left(D(\theta)\frac{\partial \theta}{\partial z}\right).\tag{2}$$

Re-writing this equation leads to the divergence-based form of the Richards equation:

$$\frac{\partial \theta}{\partial t} = \frac{\partial}{\partial z}\left[\frac{K(\theta)}{\theta}\,\theta - \frac{\partial D(\theta)}{\partial z}\,\theta\right] + \frac{\partial^2}{\partial z^2}\left(D(\theta)\,\theta\right),\tag{3}$$

where $z$ is the vertical position (positively upward) in the soil domain (m), $K$ the hydraulic conductivity (m s$^{-1}$), $D$ the water diffusivity (m² s$^{-1}$), $\Psi$ the matric potential (m), $\theta(t)$ the soil water content (m³ m$^{-3}$) and $t$ the simulation time (s).

Eq. 3 is formally equivalent to the Fokker-Planck equation (Risken, 1984). The first term of the equation
corresponds to a drift/advection term characterizing the advective downward velocity $v$ (m s$^{-1}$) of fluid fluxes driven by gravity:

$$-v(\theta) = \frac{K(\theta)}{\theta} - \frac{\partial D(\theta)}{\partial z}.\tag{4}$$

The second term of Eq. 3 represents diffusive fluxes driven by the soil moisture or matric potential gradient and controlled by diffusivity $D(\theta)$ (cf. Eq. 1). Eq. 3 can then be solved by a non-linear random walk of volumetric

water particles (Zehe and Jackisch, 2016). The non-linearity arises due to the dependence of $K$ and $D$ on soil moisture and hence the particle density. The vertical displacement of water particles is described by the Langevin equation:

$$z_i(t + \Delta t) = z_i(t) - \left( \frac{K(\theta_r + i \cdot \Delta\theta)}{\theta(t)} + \frac{\partial D(\theta_r + i \cdot \Delta\theta)}{\partial z} \right) \cdot \Delta t + Z\sqrt{2 \cdot D(\theta_r + i \cdot \Delta\theta) \cdot \Delta t} \qquad i = 1, \dots, N_B, \qquad (5)$$

where the second term describes downward advection/drift of water particles driven by gravity on basis of the hydraulic conductivity $K$ (m s$^{-1}$). The term $\frac{\partial D(\theta_r + i \cdot \Delta\theta)}{\partial z}$ corrects this drift term for the case of spatially variable diffusion and is hence added as upward velocity, contrary to the downward drift term (Roth and Hammel, 1996).

The third term of Eq. 5 describes diffusive displacement of water particles determined by the soil moisture gradient and controlled by diffusivity $D(\theta)$ (m s$^{-1}$) in combination with the random walk concept. Here, the expression $(\theta_r + i \cdot \Delta\theta)$ represents the aforementioned fraction of the actual soil water content $\theta(t)$ (cf. Sect. 2.1) that is stored in a certain pore size of the soil domain. Note that $i$ is the number of a bin of $N_B$ total bins representing the certain pore size in which the particle is stored, $\theta_r$ the residual soil moisture, $\Delta\theta$ the size/water content range of a bin, and

$Z$ a random number from a standard normal distribution.

*Model assumptions*

The above described distribution of water particle displacements to different pore sizes/bins ("binning") was the key to simulating soil water dynamics in the case of pure matrix flow, in agreement with the Richards equation

and field observations (Zehe and Jackisch, 2016). This binning of particle displacements is defined by the water diffusivity and hydraulic conductivity curve. These curves are separated into $N_B$ bins, using a step size of $\Delta\theta = \frac{(\theta(t) - \theta_r)}{N_B}$ from the residual moisture $\theta_r$ to the actual moisture $\theta(t)$ (Fig. 1). Zehe and Jackisch (2016) found that 800 bins are sufficient to resolve both curves. This particle binning concept enables also the simulation of non-equilibrium conditions in the water infiltration process. To that end, a second type of particles (event particles) is

introduced to treat infiltrating event water. These particles initially travel, purely by gravity, in the largest pores and experience a slow mixing with pre-event particles in the soil matrix during a characteristic mixing time. This non-equilibrium flow in the matrix is laminar, as Eq. 5 is based on the theory of the Richards equation (Eq. 1). An adaptive time stepping is used to fulfil the Courant criterion to ensure that particles do not travel farther than the length of a grid element $dz$ in a time step.

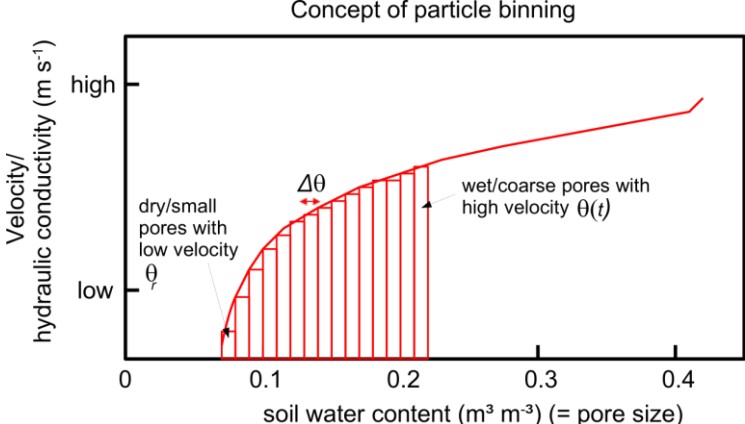

**Figure 1.** Particle binning concept. All particles within an element of the Euler-Grid are distributed to bins (= red rectangles) representing fractions of the actual soil water content stored in different pore sizes. Displacements of these particle fractions are determined by the corresponding flow velocities and diffusivities (figure taken from Sternagel et al. (2019)).

**2.2.2 Transport of conservative solutes and the macropore domain**

In our previous work (Sternagel et al., 2019), we extended the scope of the Lagrangian approach (i) to account for simulations of water and solute transport in soils as well as (ii) by a structural macropore/preferential flow domain, and included both extensions into the LAST-Model. We tested this extended approach using bromide tracer and macropore data of plot-scale irrigation experiments at four study sites and compared it to simulations of HYDRUS

1-D. At two sites dominated by well-mixed matrix flow, both models showed equal performance but at two preferential flow dominated sites, LAST performed better. We refer to Sternagel et al. (2019) for additional details on the model and results.

*Solute transport*

Each water particle is characterized by its position in the soil domain, water mass and a solute concentration. This means that there is no second species of particles representing solutes. Each water particle is tagged by a solute mass that is defined by the product of solute concentration and water particle volume. Hence, we do not use a separate, specific equation for the transport of solutes in LAST. Solutes are displaced together with the water particles according to the varying particle displacements defined by Eq. 5. Subsequent to the displacement,

diffusive mixing and redistribution of solutes among all water particles in an element of the Euler-Grid is calculated by summing their solute masses and dividing this total mass amount by the number of water particles present. Due to this perfect solute mixing process, the solute mass carried by a water particle may vary in space and time. In this context, it is important to recall that the use of an Euler-Grid to calculate soil water contents and solute concentrations in Lagrangian models may lead to the problem of artificial over-mixing (e.g. Boso et al., 2013; Cui

et al., 2014; e.g. Berkowitz et al., 2016). This is because water and solutes are assumed to mix perfectly within the elements of the Euler-Grid, which may lead to a smoothing of gradients in the case of coarse grid sizes. This might lead to overestimates of concentration dilution while solutes infiltrate into and distribute within the soil domain (Green et al., 2002, cf. Sect. 6.2).

*Macropore domain*

LAST offers a structured preferential flow domain consisting of a certain number of macropores (Fig. 2a). Macropores are classified into the three depth classes - deep, medium or shallow - to reflect the corresponding

variations of macropore depths observed at a study site. With this approach, we may account for a depth-dependent exchange of water and solutes between the matrix and macropore domains. The parameterization of the preferential flow domain may hence largely rely on observable field data, such as the number of macropores of certain diameters, their length distribution, and hydraulic properties. When such field observations are not available, the

parameters can be estimated by inverse modelling using tracer data. The actual water content and the flux densities of the topsoil control infiltration and distribution of water particles to both domains. The soil water content determines the matric potential and hydraulic conductivity of the soil matrix, while flow in macropores is controlled by friction and gravity. After the infiltration, macropores gradually fill from the bottom to the top by assuming purely gravity-driven, advective flow in the macropore domain (Fig. 2b). Interactions among macropores

and matrix are represented by diffusive mixing and exchange of water and solutes between both flow domains, which depends also on the matric potential and water content (Fig. 2c).

We provide a detailed description of Fig. 2 with the structure of the macropore domain, infiltration and filling of macropores as well as exchange processes between macropores and matrix in the Appendix.

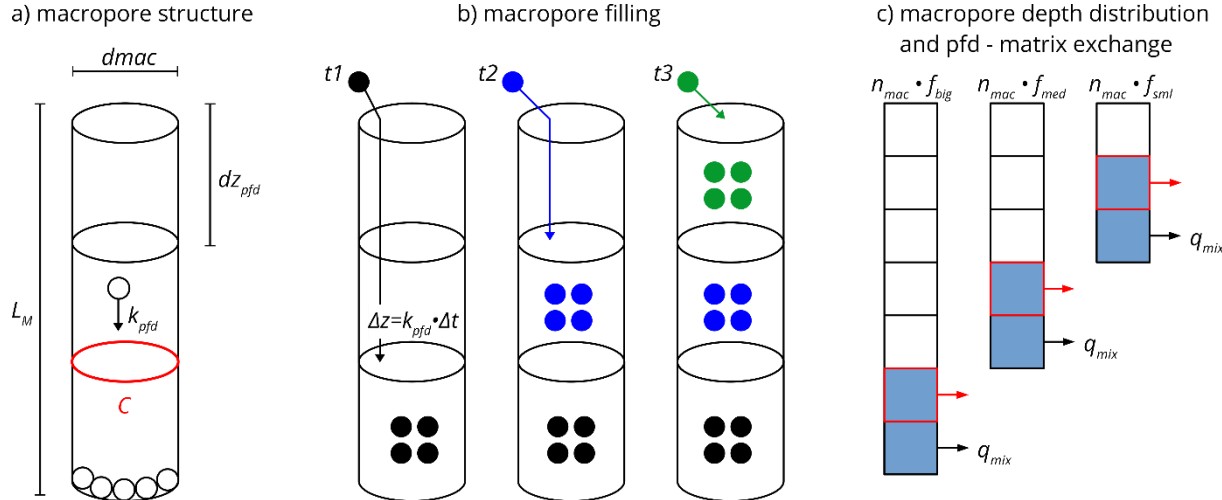

**Figure 2.** Conceptual visualization of **a)** the structure of a single macropore, **b)** the macropore filling with gradual saturation of grid elements, exemplarily shown for three points in time (t1-t3), whereby at each time new particles (differently coloured related to the current time) infiltrate the macropore and travel into the deepest unsaturated grid element, and **c)** the macropore depth distribution and diffusive mixing of water from saturated parts of macropores (blue filled squares) into the matrix (cf.

Sect. 2.2.2). The figure was adapted from Sternagel et al. (2019).

## 3 Concept and implementation of reactive solute transport into the LAST-Model

The main objective of this study is to present a method to simulate fluid flow with reactive solute transport in structured, partially saturated soils, using a Lagrangian perspective. The method is illustrated through the implementation of a routine into the LAST-Model, to simulate the movement of reactive substances through the

soil zone under the influence of sorption and degradation processes (Fig. 3). This is achieved by assigning an additional reactive solute concentration $C_{rs}$ (kg m$^{-3}$) to each water particle. A water particle can hence carry a reactive solute mass $m_{rs}$ (kg), which is equal to the product of reactive solute concentration and its water volume. Transport and mixing of the reactive solute masses within a time step are simulated in the same way as for the conservative solute (cf. Sect. 2.2.2) (Sternagel et al., 2019). After the solute mixing and mass redistribution among

water particles, the reactive solute mass of each particle can change due to a non-linear mass transfer (adsorption,

desorption) between water particles and the sorption sites of the adsorbing solid phase, which are determined by the substance-specific and site-specific Freundlich isotherms (cf. Sect. 3.1). The adsorbed reactive solute mass in the soil solid phase can then be reduced by degradation following first-order kinetics driven by the half-life of the substance (cf. Sect. 3.2). These two reactive solute processes take place in the soil matrix as well as in the wetted parts of the macropores, and their intensity can vary with soil depth as detailed in the following sections.

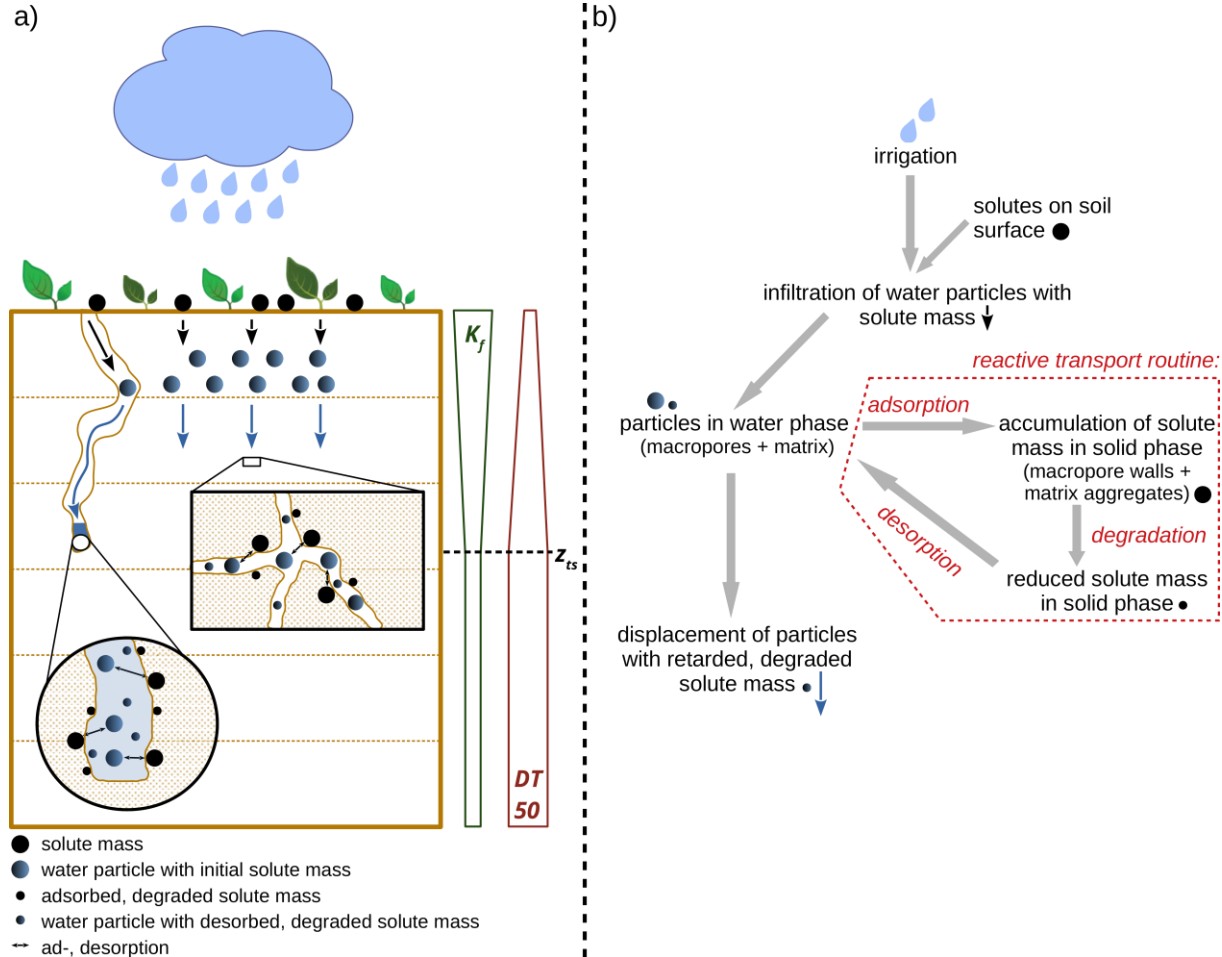

**Figure 3. a)** Overview sketch of sorption and degradation processes in the soil domain. Down to the predefined depth $z_{ts}$ (m), we assume the topsoil with linearly-decreasing $K_f$ and linearly-increasing $DT50$ values to account for the depth-dependence of sorption and degradation, respectively. Below $z_{ts}$ in the subsoil, we assume constant values. **b)** Flow chart to illustrate the sequence of reactive solute transport. The pictograms of the sketch are assigned to the respective positions and steps of the flow chart.

## 3.1 Retardation of solute transport via non-linear sorption between water and solid phase

### 3.1.1 Implementation of retardation

The interplay of adsorption and desorption characterizes the retardation process and implies that the transport velocity of a reactive solute is smaller than the fluid velocity. This is commonly represented by reducing the solute transport velocity by a retardation factor. This retardation factor describes the ratio between the fluid velocity and the solute transport velocity based on the slope of a sorption isotherm. However, this concept is not applicable in our framework, because solute masses are carried by the water particles, and travel hence at the same velocity as water. We thus explicitly represent sorption processes by a related, explicit transfer of solute masses between the water and soil solid phase. The mass exchange rates are variable in time, as the solute concentrations in the water and solid phase also vary between time steps. In each time step, the solute mass exchange between both phases is

calculated by using the non-linear Freundlich isotherms of the respective solute and rate equations (Eq. 6 for adsorption, Eq. 7 for desorption).

$$m_{rs}(t) = m_{rs}(t - \Delta t) - \left(K_f \, C_{rs}^{beta}\right) \left(\frac{m_p}{\rho}\right), \tag{6}$$

where $m_{rs}$ (kg) is the reactive solute mass of a particle, $K_f \left(\left[\frac{kg}{kg}\right]^{\frac{1}{beta}}\right)$ the Freundlich coefficient/constant, $C_{rs}$ (kg m$^{-3}$) the reactive solute concentration of a particle, $beta$ (-) the Freundlich exponent, $m_p$ (kg) the water mass of a particle, $\rho$ (kg m$^{-3}$) the water density, $t$ (s) the current simulation time and $\Delta t$ (s) the time step. Note that $K_f$ and $beta$ are both empirical constants that determine the shape and slope of the sorption isotherm of a respective

substance. Both are often described as dimensionless coefficients but $K_f$ can actually adopt different forms to balance the units of the equation, particularly when $beta$ is not equal to 1.

The reversed desorption of adsorbed solutes from the soil solid phase to the water particles, in the case of a reversed solute concentration gradient between water and solid phase, is equally calculated (Eq. 7). It uses the solute concentration in the sorbing solid phase $C_{rs\_solid}$ (kg m$^{-3}$), which requires the adsorbed solute mass and the volume

of the phase $V_{soil}$ (m³). In this way, the total desorbed solute mass is calculated for an entire grid element and must be divided by the present particle number $N_P$ (-) to equally distribute the desorbed solute mass among the water particles. The sorption process is hence controlled by a local concentration gradient between water and solid phase within an element of the Euler-Grid.

$$m_{rs}(t) = m_{rs}(t - \Delta t) + \frac{\left(K_f \, C_{rs\_solid}^{beta}\right) V_{soil}}{N_P} \tag{7}$$

### 3.1.2 Assumptions for the parameterization of the sorption process

Generally, sorption is a non-linear process, which reflects the limited availability of adsorption sites and hence, exchange rate limitations. This may cause imperfect sorption, which can lead to the observation of early mass arrivals and long tailings in breakthrough curves (e.g. Leistra, 1977). Thus, our approach calculates the non-linear

adsorption or desorption of solute masses, as a function of the solute concentration or loading of the sorption surfaces of the sorbent. Hence, in a given time step, the higher the solute concentration in the solid phase, the fewer the solute masses that can be additionally adsorbed from the water phase, and vice versa. In the approach developed here, the sorption process proceeds only until a concentration equilibrium between both phases is reached. At this point, there is no further adsorption or desorption of solute masses until the concentration of one phase is again

disequilibrated by, e.g., the infiltration of water into the water phase or by solute degradation in the solid phase. In the case that the concentration of a reactive solute in the water phase is higher than its solubility, the excess solute masses leave solution and are adsorbed to the soil solid phase.

With regard to pesticides, the major pesticide sorbent is soil organic matter and its quantity and quality determines

to a large fraction the soil sorption properties (Farenhorst, 2006; Sarkar et al., 2020). Several studies revealed that in the topsoil, enhanced sorption of pesticides occurs due to the often high content of organic matter, which may reflect bioavailability by an increased amount of sorption sites in the non-mineralized organic matter (e.g. Clay

and Koskinen, 2003; Jensen et al., 2004; Boivin et al., 2005; Rodríguez-Cruz et al., 2006). This implies that the conditions in the topsoil generally facilitate the sorption of dissolved solutes. While different depth profiles of the $K_f$ value could be implemented depending on available data, to account for this depth-dependence of sorption processes, we here apply a linearly-decreasing distribution of the $K_f$ value over the grid elements of the soil domain between two predefined upper and lower value limits for the topsoil. The depth of the topsoil ($z_{ts}$) can be adjusted individually and for our applications; here, we set it to 50 cm. Below this soil depth, we assume the subsoil and apply constant $K_f$ values. The exact $K_f$ parameterizations of the respective model setups at the different sites are explained in Sect. 4.2.1 and 4.2.2, and summarized in Table 2.

*Sorption in macropores*

While sorption generally controls pesticide leaching in the soil matrix, the processes are different in macropores. Sorption in macropores is often limited because the time scale of vertical advection is usually much smaller than the time required by solute molecules to diffuse to the macropore walls (Klaus et al., 2014). However, sorption may occur to a significant degree once water is stagnant in the saturated parts of the macropores (Bolduan and Zehe, 2006). This stagnancy facilitates the possibility for sorption of reactive solutes between macropore water and the macropore walls. The macropore sorption processes are also described and quantified by the Freundlich approach and Eq. 6.

## 3.2 First-order degradation of adsorbed solutes in soil solid phase

### 3.2.1 Implementation of degradation

Reactive solutes such as pesticides are commonly biodegraded and therewith transformed into metabolite/child compounds by the metabolism or co-metabolism of microbial communities that are present mainly on the surfaces of soil particles. The immobilization of a reactive substance, due to adsorption, favours degradation, when the residence time in the adsorbing solid phase is sufficiently long for metabolization. Many pesticides are subject to co-metabolic degradation, which often follows first-order kinetics and can hence be characterised by an exponential decay function

$$C_t = C_0 \, e^{-k \, t}, \tag{8}$$

where $C_t$ (kg m$^{-3}$) is the concentration of the pesticide after the time $t$ (s), $C_0$ (kg m$^{-3}$) the initial concentration and $k$ (s$^{-1}$) the degradation rate constant.

Based on the first-order kinetics of Eq. 8, we apply a mass rate equation (Eq. 9) for the degradation of adsorbed solute masses on the macroscopic scale of an element of the Euler-Grid:

$$m_{sp}(t + \Delta t) = m_{sp}(t) \left( 1 - \left( k_d \, \frac{\Delta t}{86400} \right) \right), \tag{9}$$

where $m_{sp}(t)$ and $m_{sp}(t + \Delta t)$ (kg) are the reactive solute masses in the soil solid phase of the current time step and of the next time step after degradation, and $\Delta t$ (s) the time step. The kinetics of this degradation process are

determined by the half-life *DT50* (d) of the respective substance, with the relationship between *DT50* and a daily degradation $k_d$ (d$^{-1}$) given by

$$k_d = \frac{ln(2)}{DT50}.$$   (10)

### 3.2.2 Assumptions for the parameterization of the degradation process

Turnover and degradation of pesticides depend in general on the substance-specific chemical properties and the microbial activity in soils (Holden and Fierer, 2005). Microbial activity in soil depends on many factors, including organic matter content, pH, water content, temperature, redox potential and carbon-nitrogen ratio. As these factors are usually highly heterogeneous in space, considerable research has focused on spatial differences in pesticide turnover potentials. Some of these studies determined that pesticide turnover rates typically decrease within the top meter of the soil matrix (e.g. El-Sebai et al., 2005; Bolduan and Zehe, 2006; Eilers et al., 2012). This is because the topsoil provides conditions that facilitate enhanced microbial activity (Fomsgaard, 1995; Bending et al., 2001; Bending and Rodriguez-Cruz, 2007). The simplest way to account for such a depth-dependent degradation is a linear increase of the *DT50* value from the topsoil surface to a predefined depth $z_{ts}$, which is set to 50 cm. This value is in line with the assumption of the depth-dependent $K_f$ parameter, and was estimated based on the findings of the aforementioned studies. In the subsoil below 50 cm, we apply constant *DT50* values (cf. Sect. 3.1). The exact *DT50* parameterizations of the respective model setups at the different sites are explained in Sect. 4.2.1 and 4.2.2, and summarized in Table 2.

*Degradation in macropores*

The presence of macropores allow pesticides to bypass the topsoil matrix, while they may infiltrate and thus be more persistent in the deeper subsoil matrix where the turnover potential is decreased. As biopores like worm burrows often constitute the major part of macropores in agricultural soils, a number of studies have focused on their key role for pesticide transformation (e.g. Binet et al., 2006; Liu et al., 2011; Tang et al., 2012). These studies consistently revealed an elevated bacterial abundance and activity in the immediate vicinity of worm burrows (Bundt et al., 2001; Bolduan and Zehe, 2006), comparable to the optimum conditions in topsoil. This is attributed to a positive effect of enhanced organic carbon, nutrient and oxygen supply that may lead to increased adsorption and degradation rates in macropores. Thus, we assume that degradation also takes place in the adsorbing phase of the macropores, which can be quantified with Eq. 9. We apply different $K_f$ and *DT50* values in the macropores that are in the range of the topsoil values (cf. Table 2).

### 4 Model application tests

The proposed method to simulate reactive solute transport in a Lagrangian approach is tested by using LAST to simulate irrigation experiments with conservative bromide tracer and the herbicide IPU as a representative reactive substance, at two study sites in the Weiherbach catchment (Zehe and Flühler, 2001). Here, conservative means that a solute is neither subject to sorption nor to degradation. These two sites are dominated by either matrix flow under well-mixed conditions (site 5) or preferential macropore flow (site 10) on a time scale of 2 days. These experiments are also simulated with the HYDRUS 1-D model. To test the method on simulation periods longer

than 2 days, we use data from an additional plot-scale (site P4) irrigation experiment (Klaus et al., 2014) on time scales of 7 and 21 days. Finally, we evaluate the method by simulating the breakthrough and remobilization of the herbicide Flufenacet that was observed in the tile-drain of a field site within two irrigation phases: one day and three weeks after substance application.

## 4.1 Characterization of the irrigation experiments

### Study area: The Weiherbach catchment

The Weiherbach valley extends over a total area of 6.3 km² and is located in the southwest of Germany. The land is used mainly for agriculture. The basic geological formation of the valley is characterized by Pleistocene Loess layer up to 15 m thick, which covers Triassic Muschelkalk marl and Keuper sandstone. At hillfoots, the hillslopes show a typical Loess catena with erosion-derived Colluvic Regosols while at hilltops and -mids, mainly Calcaric Regosols or Luvisols are present. More detailed information on the Weiherbach catchment is provided in Plate and Zehe (2008).

### Pesticides Isoproturon (IPU) and Flufenacet (FLU)

IPU is an herbicide, which is commonly applied in crops to control annual grasses and weeds. IPU has a moderate water solubility of 70.2 mg L$^{-1}$ and is regarded as non-persistent (mean *DT50* in field: 23 d) and moderately mobile (mean $K_f$ = 2.83) in soils (see also typical $K_f$ and *DT50* value ranges in Table 2). IPU is ranked as carcinogenic and its turnover in soils forms, mainly, the metabolite desmethylisoproturon (Lewis et al., 2016).

FLU is an herbicide that can be applied for a broad spectrum of purposes, but is used especially in combination with other herbicides to control grasses and broad-leaved weeds. FLU is regarded as moderately soluble (51 mg L$^{-1}$) and is not highly volatile (mean $K_f$ = 4.38) but may be quite persistent in soils (up to *DT50* in field: 68 d) under certain conditions. FLU is classified as moderately toxic to humans and its turnover in soils mainly forms the metabolites FOE sulphonic acid, FOE oxalate and FOE alcohol (Lewis et al., 2016).

### 4.1.1 Plot-scale experiments of Zehe and Flühler (2001) at the well-mixed site 5 and the preferential flow dominated site 10

At site 5, the soil moisture and soil properties were initially measured on a defined plot area of 1.4 m x 1.4 m. Before the irrigation, 0.5 g of IPU were applied, distributed evenly, on the surface of the plot area. After one day, the IPU loaded plot area was irrigated by a rainfall event of 10 mm h$^{-1}$ of water for 130 minutes with 0.165 g L$^{-1}$ of bromide. After another day, soil samples were taken along a vertical soil profile of 1 m x 1 m in a grid of 0.1 m x 0.1 m. Thus, ten soil samples were collected in each 10 cm depth interval down to a total depth of 1 m. In subsequent lab analyses, the IPU and bromide concentrations of all samples were measured. The soil at site 5 is a Calcaric Regosol (Working Group WRB, 2014) and flow patterns reveal a dominance of well-mixed matrix flow without considerable influence of macropore flows. This is the reason for using site 5 to evaluate our reactive solute transport approach under well-mixed flow conditions. Table 1 provides all experimental data.

The experiment at site 10 was conducted similarly with the initial application of 1.0 g of IPU on the soil plot and one day later a block rainfall of 11 mm h$^{-1}$ for 138 minutes. The soil at site 10 can be classified as Colluvic Regosol

(Working Group WRB, 2014) and shows numerous worm burrows that can facilitate preferential flow. Hence, we select study site 10 for the evaluation of our reactive solute transport approach during preferential flow conditions. The density and depth of the worm burrow systems were examined extensively at this study site. Horizontal layers in different depths of the vertical soil profile were excavated (cf. Zehe and Blöschl, 2004; van Schaik et al., 2014) and in each layer the number of macropores was counted and their diameters and depths were measured. These detailed measurements provided an extensive dataset of the macropore network. Table 1 again contains all experimental data.

### 4.1.2 Plot- and field-scale experiments of Klaus et al. (2014)

Klaus et al. (2014) conducted irrigation experiments in the Weiherbach catchment to corroborate the importance of macropore connectivity to tile drains for tracer and pesticide leaching into surface waters. A series of three irrigation experiments with bromide tracer, IPU and FLU were performed on a 20 x 20 m field site, which also included the sampling of these substances in different plot-scale soil profiles. We focus first on the plot-scale experiment in which the field site was irrigated in three individual blocks with a total precipitation sum of 34 mm over 220 minutes. Additionally, a total of 1600 g of bromide was applied on the field site. We concentrate on site P4 where soil samples were collected in a 0.1 m x 0.1 m grid down to a depth of 1 m after 7 days, and their corresponding bromide concentrations measured. Patterns of worm burrows in the first 15 cm of the soil were also examined (Table 3). The present soil is a Colluvisol (Working Group WRB, 2014) with a strong gleyic horizon present in a depth between 0.4 and 0.7 m, which causes a decreasing soil hydraulic conductivity gradient with depth that leads to almost stagnant flow conditions in the subsoil (Klaus et al., 2013). In general, the experiment design, soil sampling and data collection are similar to the experiments of Zehe and Flühler (2001). Initial soil water contents and all further experiment parameters as well as the soil properties at the field site are listed in Table 3.

Second, we focus on two other irrigation experiments of Klaus et al. (2014) on the field-scale in which FLU concentrations were measured at the outlet of a tile-drain tube. The tube drained the entire field site and was located 1-1.2 m below the surface. Before irrigation, a total of 40 g FLU was applied on the surface of the 400 m² field site. In a first irrigation phase, the field site was irrigated in three individual blocks with a total precipitation of 41 mm over 215 minutes and simultaneously, water samples were taken at the outlet of the tile-drain tube. These samples were analysed for FLU as explained in Klaus et al. (2014). After a period of three weeks, in which the field site remained untouched without further irrigation and FLU application, the field site was then again irrigated in two individual blocks with a total precipitation of 40 mm over 180 minutes and the FLU concentration in the tile-drain outflow measured. The objective was to examine the breakthrough of remobilized FLU that was previously adsorbed in soil.

The soil of the field site is again a Colluvisol (Working Group WRB, 2014). Overall, the soil exhibits two ploughed layers between 0-10 cm and 10-35 cm above a third, unaffected Colluvisol layer (Klaus and Zehe, 2010). Klaus and Zehe (2010) also found that 10 macropores/m² reaching into the depths of the tile-drain tube is a good estimate for simulations at this study site. Initial soil water contents and all further experimental parameters are listed in Table 4.

**4.2 Model setups**

To compare our 1-D simulation results to the observed 2-D concentration data of the plot-scale experiments, the latter are averaged laterally in each of the 10 cm depth intervals. Note that the corresponding observations provide solute concentration per dry mass of the soil while the LAST-Model simulates concentrations in the water phase and adsorbed solute masses in the soil solid phase, respectively. We thus compare simulated and observed solute masses and not concentrations in the respective depths. Note that the experimental parameters in Tables 1, 3 and 4 are measured data from the above-described experiments and can be used directly to parameterize the LAST-Model without fitting. In Sternagel et al. (2019), we explain in detail how the observed data are processed, particularly for the macropore domain, and explain the model sensitivity to the uncertainty range of observed data (e.g. to the saturated hydraulic conductivity).

**4.2.1 Model setups of simulations at well-mixed plot site 5**

*LAST-Model setup at well-mixed site 5*

As site 5 is dominated by well-mixed matrix flow, we deactivate the macropore domain of LAST and simulate IPU and bromide transport solely in the matrix domain at this site. Without the influence of macropores, we assume here only small penetration depths of solutes through the first top centimetres of the soil, in line with previous simulations at other well-mixed sites in the Weiherbach catchment (Sternagel et al., 2019). This means that solutes may remain in the upper part of the topsoil, so that a depth-dependent parameterization of sorption and degradation (cf. Sect. 3.1, 3.2) appears, as a first guess, not necessary at this site. Thus, we apply constant values of $K_f$ and $DT50$ (Table 2) and use mean values under field conditions for IPU from the Pesticide Properties DataBase (PPDB) (Lewis et al., 2016). Consistent with the experiments, we use a matrix discretization of 0.1 m. Initially, the soil domain contains 2 million water particles, but no solute masses. All further experiment and simulation parameters are shown in Table 1.

*HYDRUS 1-D setup at well-mixed site 5*

The simulation with HYDRUS 1-D at the well-mixed site 5 is conducted with a single porosity model (van Genuchten-Mualem) and an equilibrium model for water flow and solute transport, respectively, with the Freundlich approach for sorption and first-order degradation. At the upper domain boundary, we select atmospheric conditions with a surface layer and variable infiltration intensities. At the lower boundary, we assume free drainage conditions. In general, we use the same soil hydraulic properties, model setups, initial conditions and reactive transport parameters as for LAST (cf. Tables 1, 2).

**4.2.2 Model setups of simulations at preferential flow dominated plot site 10**

*LAST-Model setup for simulations at preferential flow dominated site 10*

We use an available, extensive macropore dataset to parameterize the macropore domain at site 10. Table 1 provides the depth distribution of the macropore network, mean macropore diameters, and the distribution factors. The study of Sternagel et al. (2019) explains in detail how the macropore domain of LAST is parameterized based on available field measurements. We vertically discretize the macropores in steps of 0.05 m and assume that they initially contain neither water particles nor solute masses. A maximum of 10,000 possible particles that can be stored in a single macropore, and hence the total possible number of particles in the entire macropore domain is

given by multiplication with the total number of macropores. The studies of Ackermann (1998) and Zehe (1999) provide further descriptions of site 10 and the macropore network.

As the heterogeneous macropore network allows a rapid bypassing of solutes, we expect a considerable penetration into different soil depths. We use depth-dependent values of $K_f$ and $DT50$ for IPU in the matrix and in the macropores to account for a depth-dependent retardation and degradation (Table 2) for the simulations at site 10. Furthermore, we here use different parameterization setups of the reactive transport routine to account for the remarkably variable value ranges of $K_f$ and $DT50$ reported in various studies (e.g. Bolduan and Zehe, 2006; Rodríguez-Cruz et al., 2006; Bending and Rodriguez-Cruz, 2007; Lewis et al., 2016). To account for the related uncertainty range of the reactive transport behaviour of IPU, we distinguish between two parameter configurations for a rather weak reactive transport of IPU and a strong reactive transport with enhanced retardation and degradation of IPU (Table 2). To evaluate solely the impact of the $K_f$ value on the model sensitivity, we furthermore perform a simulation at site 10 only with activated sorption and deactivated degradation. Table 1 provides all relevant simulation and experiment parameters.

*HYDRUS 1-D setup at preferential flow dominated site 10*

The simulation with HYDRUS 1-D at the preferential flow dominated site 10 are performed with the same model setups, soil properties, initial and boundary conditions as well as reactive transport parameters as for the simulations with LAST (cf. Tables 1, 2). In contrast, we select a dual-permeability approach for water flow ("Gerke and van Genuchten, 1993") and solute transport ("Physical Nonequilibrium") at this site. These approaches distinguish between matrix and fracture domains for water flow and solute transport. It applies the Richards equation for water flow in each domain, with domain-specific hydraulic properties. The advection-dispersion equation is used to simulate solute transport and mass transfer between both domains, including terms for reactive transport with retardation and degradation (Gerke and van Genuchten, 1993). While we apply the same soil hydraulic properties in the matrix (cf. Table 1) as for the LAST simulations, the macropore domain in HYDRUS gets a faster character with a $K_s$ value of $10^{-3}$ m s$^{-1}$. We also select the Freundlich approach for sorption processes and first-order degradation.

**4.2.3 LAST-Model setup of 7-day and 21-day simulations at plot site P4**

We perform simulations for conservative bromide tracer and reactive IPU at site P4 for periods of 7 and 21 days using the parameters in Table 3, respectively. Based on examination of the macropore network, we again derive the parameterization of the macropore domain (Table 3). In line with the LAST-Model setups in Sect. 4.2.1 and 4.2.2, we apply the same discretization of the matrix $dz$ (0.1 m) and macropore (0.05 m) domain as well as number of particles in both domains (2 million; 10 k per macropore grid element). Additionally, we perform another 7-day simulation for bromide with a finer matrix discretization $dz$ of 0.05 m. Initially, macropores and matrix contain no solute masses, and the macropores also contain no water.

For the simulation of reactive IPU transport, we again apply the weak and strong reactive transport parameterizations with the depth-dependent $K_f$ and $DT50$ values of the simulations at site 10 (cf. Table 2). Additionally, we apply here a mean reactive transport parameterization. An evaluation with observed IPU mass profiles is not possible here because robust experimental data are missing. All relevant parameters of the 7-day and 21-day simulations at P4 are listed in Table 3.

### 4.2.4 LAST-Model setup of the FLU breakthrough simulations at the field site

We perform a simulation of FLU concentrations, which migrate from the soil surface into the depth of the tile-drain tube (1 m), over the entire field site, in each of the two irrigation phases. After the first irrigation phase, we assume steady-state flow conditions, as Klaus et al. (2014) found that the flow in the tile-drain tube already approached its initial value after roughly 500 minutes. This implies hydraulic equilibrium between gravity and capillary forces and thus zero soil water flow in the period of three weeks between the first and second irrigation phase. Nevertheless, adsorption and degradation of FLU are still active and simulated using mean $K_f$ and $DT50$ values in soil (Lewis et al., 2016, cf. Table 4) during the three weeks until the second irrigation phase starts.

The parameterization of the macropore domain with the amount and depth of macropores per m² follows the recommendations of Klaus and Zehe (2010). In line with the previous LAST-Model setups, we apply the same discretization of the matrix $dz$ (0.1 m) and macropore (0.05 m) domain as well as the number of particles in both domains (2 million; 10 k per macropore grid element). Macropores and matrix again contain no solute masses, and the macropores also contain no water, initially. All further simulation parameters of the FLU breakthrough simulations are listed in Table 4.

**Table 1.** Parameters of IPU plot-scale experiments and simulations, as well as soil hydraulic parameters after van Genuchten (1980) and Mualem (1976), at sites 5 and 10. Where $K_s$ is the saturated hydraulic conductivity, $\theta_s$ the saturated soil water content, $\theta_r$ the residual soil water content, $\alpha$ the inverse of an air entry value, $n$ a quantity characterizing pore size distribution, $s$ the storage coefficient and $\rho_b$ the bulk density. Further, *mac. big*, *mac. med* and *mac. sml* describe the lengths of big, medium and small macropores as well as $f_{big}$, $f_{med}$ and $f_{sml}$ are the respective distribution factors to split the total number of macropores to these three macropore depths (cf. Sect. 2.2.2). For further details on these parameters, see Sternagel et al. (2019).

| Parameter | Site 5 | Site 10 |
|---|---|---|
| ***Experimental parameters*** | | |
| *Irrigation duration (hh:mm)* | 02:10 | 02:18 |
| *Irrigation intensity (mm h⁻¹)* | 10.70 | 11.00 |
| *Applied IPU mass (kg)* | $5 \cdot 10^{-4}$ | $1 \cdot 10^{-3}$ |
| *Recovery rate (%)* | 84.4 | 91 |
| *Initial soil moisture in 15 cm (%)* | 23.7 | 27.8 |
| *Soil type* | Calcaric Regosol | Colluvic Regosol |
| $K_s$ *(m s⁻¹)* | $1.00 \cdot 10^{-6}$ | $1.00 \cdot 10^{-6}$ |
| $\theta_s$ *(m³ m⁻³)* | 0.46 | 0.46 |
| $\theta_r$ *(m³ m⁻³)* | 0.04 | 0.04 |
| $\alpha$ *(m⁻¹)* | 4.0 | 3.0 |
| *n (-)* | 1.26 | 1.25 |
| *s (-)* | 0.38 | 0.38 |
| $\rho_b$ *(kg m⁻³)* | 1300 | 1500 |
| *Number of macropores/m² (-)* | - | 92 |
| *Mean macropore diameter (m)* | - | 0.005 |
| *mac. big (m)* | - | 0.8 |
| *mac. med (m)* | - | 0.5 |
| *mac. sml (m)* | - | 0.2 |
| $f_{big}$ *(-)* | - | 0.14 |
| $f_{med}$ *(-)* | - | 0.37 |
| $f_{sml}$ *(-)* | - | 0.49 |
| ***Simulation parameters*** | | |
| *Simulation time t (s)* | 172,800 (= 2 Days) | |
| *Time step Δt (s)* | dynamic | |

| | | |
|---|---|---|
| Particle number in matrix (-) | | 2 Mill. |
| Water mass of particle in matrix (kg) | $1.9 \cdot 10^{-4}$ | $2.2 \cdot 10^{-4}$ |
| Particle number in macropore domain (-) | - | 920 k |
| Water mass of particle in macropore domain (kg) | - | $1.6 \cdot 10^{-6}$ |

**Table 2.** Reactive transport parameters of IPU at the sites 5, 10 and P4. The upper and lower value limits in the squared brackets describe the value ranges of the depth-dependent $K_f$ and $DT50$ parameters between soil surface and the starting depth of the subsoil (cf. Fig. 3, Sect. 3.1.2, 3.2.2). At sites 10 and P4, we distinguish between two parameter configurations for a rather weak reactive transport of IPU and a strong reactive transport with enhanced retardation and degradation of IPU. Exclusively at site P4, we additionally apply a mean reactive transport parameter configuration for the 7 and 21 day simulations.

| Parameter | Site 5 | Site 10 and P4 | | P4 |
|---|---|---|---|---|
| | | weak | strong | mean |
| $K_f$ for IPU in soil matrix (-) [upper limit; lower limit] | [2.83; 2.83] | [1; 0.26] | [27; 3] | [14; 1.63] |
| DT50 for IPU in soil matrix (d) [upper limit; lower limit] | [23; 23] | [23; 44] | [3; 12] | [13; 28] |
| $K_f$ for IPU in macropores (-) | - | 5 | 10 | 7.5 |
| DT50 for IPU in macropores (d) | - | 15.6 | 10 | 12.8 |
| beta (-) | 0.8 | 0.8 | 0.8 | 0.8 |

**Table 3.** Parameters of 7-day bromide experiment at plot-scale site P4 (Klaus et al., 2014) and simulation parameters as well as soil hydraulic parameters after van Genuchten (1980) and Mualem (1976). For parameter definitions and further details on these parameters, see caption of Table 1 and Sternagel et al. (2019). Note that only one macropore depth of 15 cm was observed at this site.

| Parameter | P4 |
|---|---|
| ***Experimental parameters*** | |
| Irrigation duration (hh:mm) | 03:40 |
| Total irrigation sum (mm) | 34.00 |
| Applied bromide mass on 400 m² field site (kg) | 1.6 |
| Recovery rate (%) | ~ 100 |
| Initial soil moisture in 10 cm (%) | 24.8 |
| Initial soil moisture in 20 cm (%) | 27.1 |
| Initial soil moisture in 30 cm (%) | 27.0 |
| Initial soil moisture in 40 cm (%) | 28.44 |
| Initial soil moisture in 60 cm (%) | 33.11 |
| Initial soil moisture in 100 cm (%) | 29.60 |
| Soil type | Colluvisol |
| $K_s$ in: topsoil; gleyic horizon (subsoil) (m s$^{-1}$) | $1.00 \cdot 10^{-5}$; $1.00 \cdot 10^{-8}$ |
| $\theta_s$ in: topsoil; gleyic horizon (subsoil) (m³ m$^{-3}$) | 0.50; 0.4 |
| $\theta_r$ in: topsoil; gleyic horizon (subsoil) (m³ m$^{-3}$) | 0.04; 0.11 |
| $\alpha$ in: topsoil; gleyic horizon (subsoil) (m$^{-1}$) | 1.9; 3.8 |
| $n$ in: topsoil; gleyic horizon (subsoil) (-) | 1.25; 1.20 |
| s (-) | 0.38 |
| $\rho_b$ (kg m$^{-3}$) | 1500 |
| Number of macropores/m² (-) | 68 |
| Mean macropore diameter (m) | 0.003 |
| mac. big (m) | 0.20 |
| mac. med (m) | - |
| mac. sml (m) | - |

| | |
|---|---|
| $f_{big}$ (-) | 1.0 |
| $f_{med}$ (-) | - |
| $f_{sml}$ (-) | - |
| **Simulation parameters** | |
| Simulation time $t$ (s) | 604,800 (= 7 days) |
| Time step $\Delta t$ (s) | dynamic |
| Particle number in matrix (-) | 2 Mill. |
| Water mass of particle in matrix (kg) | $2.3 \cdot 10^{-4}$ |
| Particle number in macropore domain (-) | 680 k |
| Water mass of particle in macropore domain (kg) | $1.4 \cdot 10^{-7}$ |

**Table 4.** Parameters of field-scale FLU breakthrough experiment (Klaus et al., 2014) and simulation parameters as well as soil hydraulic parameters after van Genuchten (1980) and Mualem (1976). For parameter definitions and further details on these parameters, see caption of Table 1 and Sternagel et al. (2019). Note that only one macropore depth of 1 m reaching the depth of the tile-drain tube is applied.

| Parameter | Field site |
|---|---|
| **Experimental parameters** | |
| Irrigation duration (hh:mm) of 1. and 2. irrigation phase | 03:35; 02:00 |
| Total irrigation sum (mm) of 1. and 2. irrigation phase | 41.00; 40.00 |
| Applied FLU mass on field site (kg) | 0.04 |
| Initial mean soil moisture (%) | 28.0 |
| Soil type | Colluvisol |
| $K_s$ in: 0-10 cm; 10-35 cm; below 35 cm (m s$^{-1}$) | $5.00 \cdot 10^{-4}$; $2.70 \cdot 10^{-5}$; $5.00 \cdot 10^{-5}$ |
| $\theta_s$ in: 0-10 cm; 10-35 cm; below 35 cm (m³ m$^{-3}$) | 0.46; 0.43; 0.4 |
| $\theta_r$ in: 0-10 cm; 10-35 cm; below 35 cm ) (m³ m$^{-3}$) | 0.1; 0.11; 0.04 |
| $\alpha$ in: 0-10 cm; 10-35 cm; below 35 cm ) (m$^{-1}$) | 2.4; 3.8; 1.9 |
| $n$ in: 0-10 cm; 10-35 cm; below 35 cm (-) | 1.22; 1.2; 1.25 |
| $s$ (-) | 0.38 |
| $\rho_b$ (kg m$^{-3}$) | 1500 |
| Number of macropores/m² (-) | 10 |
| Mean macropore diameter (m) | 0.005 |
| mac. big (m) | 1.00 |
| mac. med (m) | - |
| mac. sml (m) | - |
| $f_{big}$ (-) | 1.0 |
| $f_{med}$ (-) | - |
| $f_{sml}$ (-) | - |
| $K_f$ (-) for FLU in soil matrix | 4.83 |
| DT50 (d) for FLU in soil matrix | 54 |
| $K_f$ (-) for FLU in macropores | 4 |
| DT50 (d) for FLU in macropores | 19 |
| beta (-) | 0.92 |
| **Simulation parameters** | |
| Simulation time $t$ (s) | 1,814,400 (= 3 weeks) |
| Time step $\Delta t$ (s) | dynamic |
| Particle number in matrix (-) | 2 Mill. |
| Water mass of particle in matrix (kg) | $2.9 \cdot 10^{-4}$ |
| Particle number in macropore domain (-) | 680 k |
| Water mass of particle in macropore domain (kg) | $1.96 \cdot 10^{-6}$ |

**5 Results**

In the following sections, we present simulated vertical mass profiles of bromide and IPU at the different plot-scale study sites 5, 10 and P4 as well as breakthrough time series of FLU concentrations at the field site (cf. Sect. 4.1).

**5.1 Simulation results at the well-mixed plot site 5 after 2 days**

**5.1.1 IPU transport simulated with LAST**

In Fig. 4a, the reference simulation treating IPU as conservative (red profile) overestimates the transport of IPU into soil depths lower than 10 cm, with a maximum penetration depth of 40 cm. This leads in turn to simultaneous underestimation of masses in shallow depths near the soil surface (RMSE: 0.064 g, 12.8 % of applied mass). In

the case of the simulation with retardation and no degradation (yellow profile), the simulated mass profile matches the observed profile in the first 10 cm, because retardation causes mass accumulation. With additional degradation (light blue profile), the solute masses in the first 10 cm are then slightly reduced. The influence of degradation is relatively small, due to the moderate *DT50* value of 23 days and the short simulation period of two days, but it is nevertheless detectable. Overall, we find that there are indeed noticeable differences (RMSE difference of 7.3 %)

between the IPU profiles of the conservative, reference simulation and the reactive transport simulation with retardation and degradation, which is also in better accord with the observed mass profile, reflected by a smaller RMSE value of 0.027 g (5.5 % of applied mass). At the end of the simulated period of two days, a total IPU mass of 0.014 g is degraded, while the observed profile has a mass deficit of 0.078 g corresponding to a recovery rate of 84 %. This observed mass deficit cannot be explained exclusively by degradation. It might be the result of

additional mass losses in the experiment execution and lab analyses.

**5.1.2 IPU transport simulated with HYDRUS 1-D**

The IPU mass profile simulated with HYDRUS 1-D (Fig. 4b), with activated reactive transport, show similar mass patterns compared to LAST and the observed profile with a RMSE value of 0.036 g (7.3 % of applied mass). While HYDRUS overestimates the IPU masses at the soil surface, considering a stronger retardation compared to the

observation and the LAST results, it simulates the observed masses in 10-20 cm soil depth quite well. In these depths, LAST overestimates masses with a maximum penetration depth of 30 cm, which is 10 cm deeper than observed. Overall, the results of HYDRUS and LAST are in comparable agreement with the observed profile. HYDRUS simulates a total, degraded IPU mass of 0.017 g, which is in range with the LAST results (cf. Sect. 5.1.1). This means that in both models, the total degradation is similar but the distribution of the remaining IPU

masses over the soil profile differs.

**5.1.3 Bromide transport simulated with LAST**

Bromide slightly percolates into greater depths during the short-term irrigation experiment (Fig. 4c) compared to the retarded and degraded IPU (cf. Fig. 4a). The results generally underline that the Lagrangian approach is able to simulate conservative solute transport under well-mixed conditions, as we have already shown in our previous

study (Sternagel et al., 2019). The results further show that the approach is capable of treating both conservative tracers and reactive substances.

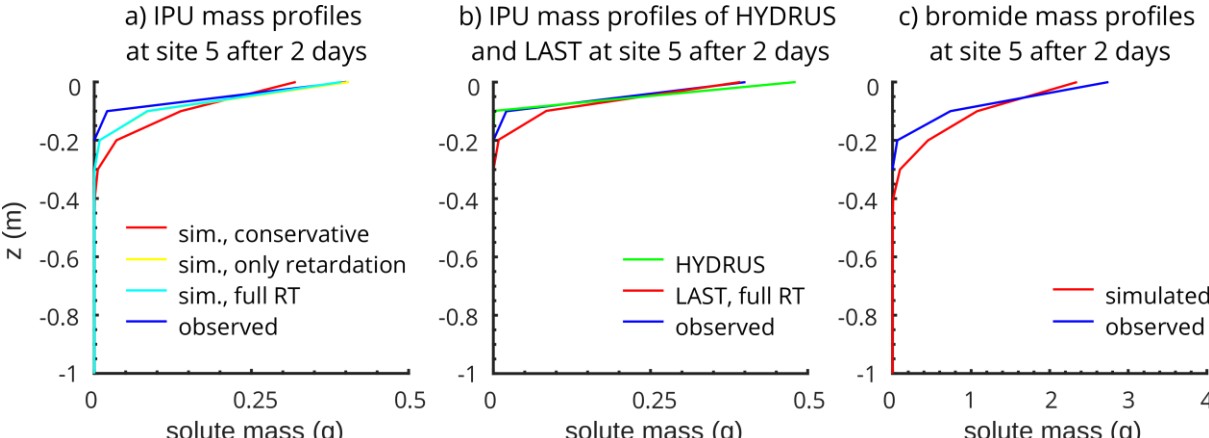

**Figure 4. a)** LAST-Model results of reactive IPU transport simulations at the well-mixed site 5 after 2 days with regarding IPU as conservative without reactive transport which serves as reference (red profile), with activated retardation but without degradation to exclusively show the effect of the sorption processes (yellow profile) and with fully activated reactive transport (light blue profile). **b)** Comparison with HYDRUS 1-D results and **c)** exemplary results of a conservative simulation with LAST for bromide.

### 5.2 Simulation results at the preferential flow dominated plot site 10 after 2 days

### 5.2.1 IPU transport simulated with LAST

Fig. 5a and 5b present results of different simulation setups compared to the observed IPU mass profile at site 10 after 2 days. Both figures comprise the observed profile as well as a profile of a reference simulation treating IPU as conservative. Fig. 5a focuses on the mass profiles resulting from simulations only with activated retardation, using low and high $K_f$ values. Fig. 5b shows results for simulations performed with full reactive transport subject to retardation and degradation, comparing parameterizations for weak and strong reactive transport. The shaded area between these profiles represents the corresponding uncertainty ranges.

In general, the typical "fingerprint" of preferential flow through macropores is clearly visible in the observed IPU mass profile. The observed mass accumulations and peaks fit well to the observed macropore depth distribution (cf. Table 1), which implies that water and IPU travelled through the macropores and infiltrated into the matrix in the respective soil depths where the macropores end. The observed mass profile shows a strong accumulation of IPU masses in depths between 70-90 cm, which cannot be explained by the relatively low number of macropores (~ 13) in this depth. One reason for this could be particle-bound transport of IPU at this study site, as proposed by Zehe and Flühler (2001). They suggested that IPU is adsorbed to mobile soil particles or colloids at the soil surface, which then travel rapidly through macropores into greater depths at this site. In comparison, the simulated conservative reference profile depicts the observed mass distribution quite well on average, although less well the heterogeneous profile shape with a RMSE value of 0.076 g (7.6 % of applied mass). The mass peaks in the depths of the macropore ends (cf. Table 1) are relatively weak, because solute masses leaving the macropores are not retarded in the matrix but instead flushed out by the water flow into deeper soil depths, resulting in a smoothed mass profile. In the surface-near soil depths between 0-10 cm, the conservative reference simulation clearly overestimates the IPU masses.

The range of simulated mass profiles, corresponding to the weak and strong reactive transport parameterization (Fig. 5b), matches the observed profile in terms of both mass amounts and shape, with a RMSE value of 0.038 g (3.8 % of applied mass). Hence, at this site, LAST performs better with activated reactive transport compared to

the conservative reference setup. The mass accumulations, which are detectable in those depths where the macropores end, arise from adsorption and retardation of solutes that infiltrated the soil matrix out of the macropores. While the simulation with full reactive transport also overestimates the IPU masses in the upper 10 cm, the simulated and observed mass profiles coincide well in the lower depths. The observed mass peak at 70-90 cm cannot be reproduced completely. Furthermore, the wider ranges between the simulated profiles for weak and strong reactive transport in the topsoil reflect the depth-dependent parameterization of especially the *DT50* values. The higher IPU amounts and lower *DT50* values in the topsoil cause a faster degradation than in underlying soil depths. In subsoil, degradation is slower and uniform, and due to smaller IPU amounts, there is no difference between the two parameterizations. The total degraded IPU masses are between 0.026 g and 0.131 g for the weak and strong reactive transport simulations, respectively. With an IPU input amount of 1 g, up to 13 % of IPU is degraded in just two days. This shows the relevance of the degradation process, even on these short time scales. The relatively high observed IPU mass recovery of 0.908 g (~ 91 %) implies a possible degraded total mass of 0.092 g, which is consistent with our simulated range.

The simulation only with retardation (Fig. 5a) reveals hardly any sensitivity to the variation of $K_f$ values. This implies that the amounts of adsorbed masses are almost equal for different $K_f$ values at the end of the simulation, and thus independent of the $K_f$ value. This might be due to non-linear adsorption, which establishes an equilibrium state between water and the adsorbing phase after a certain time (cf. Sect. 3.1). Hence, independent of the magnitude of $K_f$, no further adsorption occurs unless degradation is activated, which would lead to mass loss in the soil solid phase and a renewed adsorption capacity (cf. Fig. 5b). Higher $K_f$ values lead only to a shorter time to reach this equilibrium state; the final adsorbed masses are similar for different $K_f$ values due to the inactivated degradation in this special case.

### 5.2.2 IPU transport simulated with HYDRUS 1-D

In contrast to the findings at site 5, IPU mass profiles simulated with the dual-permeability approach of HYDRUS 1-D (Fig. 5c) do not match the observed profile at site 10, resulting in a RMSE value of 0.079 g (7.9 % of applied mass). In the first 35 cm, HYDRUS simulates a strong retardation and overestimation of masses with a maximum penetration depth of only 50 cm. In comparison, simulations with the Lagrangian approach in the LAST-Model match the observed profile better (cf. Sect. 5.2.1). However, the total degraded IPU masses of 0.028 g and 0.183 g resulting from the weak and strong reactive transport parameterizations simulated with HYDRUS are similar to those resulting from LAST simulations.

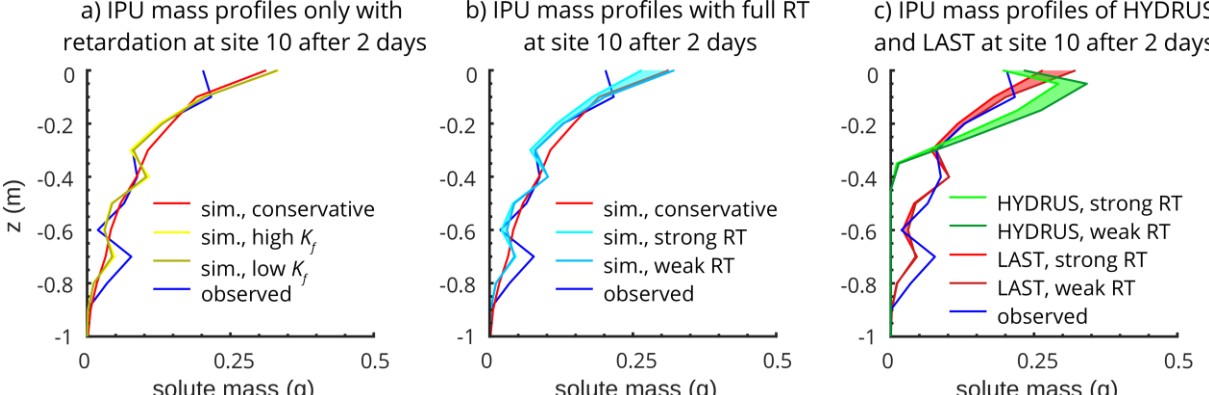

**Figure 5.** LAST-Model results of reactive IPU transport simulations at the preferential flow dominated site 10 after 2 days. **a)** The simulation is performed only with active retardation and with low and high values for $K_f$. **b)** The simulation is performed with activated retardation and degradation. The shaded area between the profiles with weak and strong reactive transport (cf. Table 2) shows the uncertainty area of the empirical $K_f$ and $DT50$ values. **c)** Comparison with HYDRUS 1-D simulation results.

### 5.3 Simulation results of LAST at plot site P4

#### 5.3.1 Bromide transport in 7 days

Fig. 6a shows the simulated and observed bromide mass profiles at site P4 after 7 days. Note that a model evaluation directly at the surface is not meaningful, because the soil sampling in the experiment started at a depth of 5 cm.

The observed mass profile is characterized by two distinct mass peaks. One peak, at 15-30 cm, probably originates from solute masses entering the matrix from the macropores in 15 cm depth, which are subsequently displaced by water movement into this depth range within the 7 days. The second peak in a depth around 60-70 cm likely originates from an accumulation of water and solutes above the less permeable gleyic horizon in this depth. In comparison, the simulated bromide mass profile simulated with a finder discretization $dz$ of 0.1 m (red solid profile) is generally shifted to greater depths, although the shape corresponds quite well to the observed profile. Between 5-30 cm, the simulated masses are underestimated, and conversely, they are overestimated between 30-55 cm depth. Obviously, LAST simulates a solute displacement that is too strong and fast into these deeper soil depths ("deep shift"), compared to the observed mass accumulation (15-30 cm), after solute masses leave the macropores and enter the matrix in 15 cm depth. The simulated mass accumulation in the gley horizon coincides quite well with the observed data, but with long tailing. Despite the almost stagnant conditions in the gley horizon, we simulate a too strong displacement of solutes into soil depths even deeper than 1 m (Fig. 6a) with the setup in LAST. This behaviour is also visible in the simulated bromide mass profile with a refined $dz$ of 0.05 m (red dashed line), but the effect is less pronounced as more bromide remains in the upper 30 cm.

#### 5.3.2 IPU transport in 7 and 21 days

Fig. 6b shows simulated IPU mass profiles after 7 days, for 6.3 g of IPU initially applied to the soil surface. These results provide insights to a possible temporal development of IPU leaching for different reactive transport parameterizations. Note that comparable observed data are unavailable (cf. Sect. 4.2.3, Table 2). The depth transport of IPU is limited compared to bromide and the reference simulation, treating IPU as non-reactive

conservative solute (red profile), which reflects IPU retardation and degradation in the topsoil. However, we observe two clear mass peaks at the end of the 15 cm deep macropores and above the gley horizon, as for bromide. In topsoil, the range between the two profiles simulated with a weak and strong reactive transport parameterization is largest. This is caused by the enhanced retardation and degradation potential and the high amount of IPU in the

topsoil. The decreased potential for sorption and degradation in the subsoil leads to negligible differences between the profiles in greater soil depths. In total, the degraded IPU masses for the two parameterizations lie between 0.514 g and 2.618 g.

After 21 days, the resulting IPU mass profiles show remarkable differences compared to the profiles after 7 days

(Fig. 6c). We observe a deeper penetration and greater range of profiles simulated with the weak and strong RT parameterization in subsoil. The mass peaks are barely detectable and mostly smoothed out along the 15 cm deep macropores and in the depth of the gley horizon. The total degraded IPU masses for the strong and weak reactive transport parameterizations range between 1.345 g and 4.625 g. Furthermore, despite applying mean reactive transport parameters, the resulting IPU mass profile (black profile) is not centred in the light blue shaded profile

range due to the non-linear character of sorption and degradation.

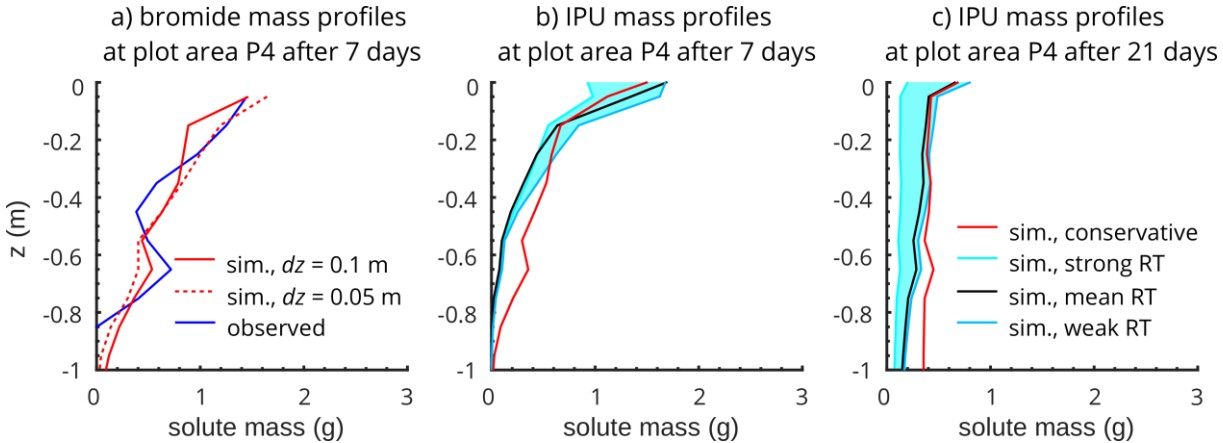

**Figure 6. a)** LAST-Model results of 7-day simulation with bromide and different soil domain discretization *dz* at the site P4. **b and c)** Show hypothetical results of a 7-day and 21-day simulation with a weak and strong reactive transport configuration

for IPU (cf. Table 2, Sect. 4.2.3) at the site P4. The bright blue range between the two profiles demonstrates a hypothetical and possible range of IPU mass profiles under the influence of retardation and degradation during 7 and 21 days at this site, respectively. The black profiles additionally show results with a mean reactive transport configuration (cf. Table 2).

**5.4 Simulation results of FLU breakthrough on field site**

During the first irrigation phase, the simulation shows deficiencies to reproduce the observed temporal dynamics and peaks of FLU concentrations (Fig. 7a). The first breakthrough peak, probably originating from FLU bypassing the matrix through macropores, is simulated after 50 minutes with a subsequently slight decrease. After 100 minutes, the simulation shows a steady increase of FLU concentrations due to delayed breakthrough of FLU

through the soil matrix. The clear underestimation of the second concentration peak after ~280 minutes can be partly explained by the fact that about 20 % of FLU was subject to rapid particle-bound transport (Klaus et al., 2014). This mechanism is not considered in our approach.

The simulation of FLU remobilization during the second irrigation phase reveals similar results. The simulated remobilization is too early (75 minutes) and followed by a second peak after 175 minutes. Both peaks originate again from a first breakthrough of FLU through macropores and subsequent leaching through the matrix (Fig. 7b). Nevertheless, the results underpin that the presented reactive transport method within the Lagrangian approach is able to reproduce the remobilization of FLU into the tile drain during the second irrigation. This implies that the approach was also capable to estimate properly the adsorption and degradation of FLU during the three weeks. Despite the limited match with the observed temporal changes of FLU concentrations, the amount of cumulated FLU masses is at the end of both irrigation phases in good agreement with the observations (Fig. 7c and 7d).

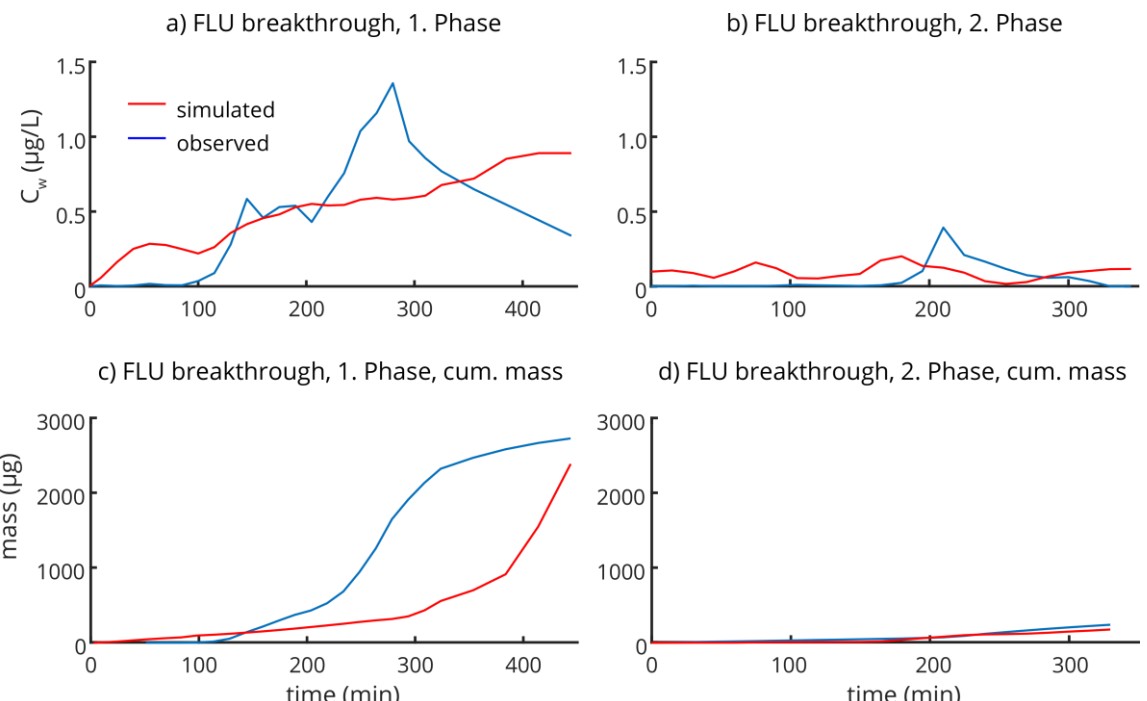

**Figure 7.** LAST-Model results of FLU breakthrough simulations with FLU concentrations $C_w$ in the tile-drain tube of the field site. **a and b)** show FLU concentration changes over time and **c and d)** present cumulated FLU masses in the two irrigation phases, respectively (cf. Sect. 4.1.2 and 4.2.4).

## 6 Discussion

The key innovation of this study is a method to simulate reactive solute transport in the vadose zone within a Lagrangian framework. In this context, we extend the LAST-Model (Sternagel et al., 2019) with the presented method for reactive solute transport to account for non-linear sorption and first-order degradation processes during transport of reactive substances such as pesticides through a partially saturated soil matrix domain and macropores. For the sorption process, we develop an explicit mass transfer approach based on the Freundlich isotherms because the usual method of using a retardation factor (cf. Sect. 3.1.1) is not applicable in the particle-based approach of LAST. Model evaluations with data from irrigation experiments, that examined plot-scale leaching of bromide and IPU under different flow conditions on various time scales as well as FLU breakthrough on a field site, corroborate the suitability of the approach and its physically valid implementation. Comparisons to simulations with HYDRUS

1-D reveal furthermore that an explicit representation of macropores and their depth distribution is favourable to predict preferential transport of solutes.

### 6.1 Sorption and degradation in the Lagrangian framework

### 6.1.1 Reactive transport under well-mixed conditions

The 2-day simulations of IPU transport at the well-mixed site 5 corroborate the validity of the Lagrangian approach and the proposed method for sorption and degradation, implemented into the Lagrangian framework of LAST (Fig. 4a). Adsorption causes an expected accumulation of IPU in topsoil layers (0-10 cm) and, consequently, reduced percolation into greater soil depths. Although degradation has only a small impact at this short time scale, due to the moderate *DT50* value of 23 days, we nevertheless observe a total degradation of 0.014 g IPU in two
days that occurs especially in the shallow soil areas where IPU accumulates. The mass profile simulated with retardation and degradation is more consistent with the observations than the reference simulation treating IPU as a conservative solute. Such a fast reaction and degradation of IPU in the topsoil can be explained by its general non-persistent and moderately mobile character, as well as an obviously very short duration of a lag-phase near the soil surface, which was also discovered by several field studies (e.g. Bending et al., 2001; Bending et al., 2003;
Rodríguez-Cruz et al., 2006). The simulated mass profiles of the benchmark simulations with the commonly used HYDRUS 1-D model are also in accord with the observations and corroborate our results; in particular, the total degraded masses are in a similar range (Fig. 4c, Sect. 5.1). This suggests that the developed reactive transport method in our Lagrangian approach performs similarly to that implemented in HYDRUS 1-D at this well-mixed site 5. This finding is in line with our previous study, which revealed that both approaches yielded similar
simulations of conservative tracer transport at matrix flow dominated sites (cf. Sternagel et al., 2019).

### 6.1.2 Impact of the macropore domain on reactive transport under preferential flow conditions

The simulation results at the preferential flow dominated site 10 show that the Lagrangian approach is capable of reproducing the observed heterogeneous IPU mass profile. The implemented depth-dependent sorption and degradation processes are particularly helpful in this context (cf. Fig. 5). However, in the entire context of this
study, it should be recognized that mean values of $K_f$ and *DT50* (cf. Table 2, Sect. 4.2.1) from the PPDB Database were determined empirically at other field sites. Measurements of these variables are laborious and not straightforward, as controls on sorption and degradation vary in space and time (Dechesne et al., 2014). The use of literature values for these parameters introduces considerable uncertainty into pesticide fate modelling (Dubus et al., 2003). We explore this uncertainty by varying the $K_f$ and *DT50* values for IPU at site 10 in the ranges
provided by the PPDB Database and further literature (cf. Table 2, Sect. 4.2.2).

The results corroborate the importance of a structural representation of macropores and their depth distribution, as implemented in the LAST-Model, consistent with the results of Sternagel et al. (2019). This is also reflected by the fact that the simulated IPU masses in the topsoil between 10-30 cm, and particularly the mass accumulations
at the depths where macropores end (cf. Table 1), match the observations, compared to the reference simulation treating IPU as conservative tracer and to simulations with HYDRUS 1-D. We conclude that an explicit representation of the macropore system with its connectivity, diameter and depth distribution, and treatment of

macropore flow and exchange with the matrix, is crucial to reproduce solute bypassing of the topsoil matrix and subsequent infiltration into the subsoil matrix.

HYDRUS 1-D does not match the heterogeneous shape of the observed mass profile at site 10, despite the use of a dual-permeability approach and the same parameterization as LAST. HYDRUS 1-D barely accounts for IPU bypassing and breakthrough to greater depths and it overestimates retardation in topsoil, which results in a high mass accumulation in the first 10 cm of soil. The total degraded IPU masses are similar in both models, and in accord with the observed data, as both models rely on first-order degradation (Gerke and van Genuchten, 1993). These results hence corroborate the findings of Sternagel et al. (2019), who concluded that HYDRUS is effective under well-mixed conditions but is limited in terms of simulating preferential flow and partial mixing between matrix and macropore flow regimes (e.g. Beven and Germann, 1982; Šimůnek et al., 2003; Beven and Germann, 2013; Sternagel et al., 2019). We propose that incorporating a similarly structured macropore domain into HYDRUS would likely improve simulations under such conditions.

However, all simulated IPU mass profiles at site 10 overestimate the observed masses within the upper 10 cm of the soil. This may be due to an additional photochemical degradation at the soil surface, surface losses due to volatilization, or even plant uptake (Fomsgaard, 1995). Such processes are difficult to detect and parameterize. A possible reason for the mismatch of the observed and simulated mass profiles in 70 cm soil depth at site 10 could be a facilitated pesticide displacement due to particle-bound transport (Villholth et al., 2000; de Jonge et al., 2004).

### 6.1.3 Sensitivity to variations of sorption and degradation parameters

The ranges of the $K_f$ and $DT50$ values for the case of a weak and strong reactive transport parameterization cause differences in the resulting mass profiles (cf. shaded areas in Fig. 5). These differences are generally stronger in topsoil and gradually decrease with depth. This is because (i) sorption and degradation rates are, due to the higher IPU masses, larger in topsoil than in the subsoil, and (ii) due to the depth-dependent parameterization (cf. Sect. 3). The results of the simulation with only sorption, and no degradation (Fig. 5a), suggest a moderate to low model sensitivity to the $K_f$ parameter. This may be due to the establishment of an equilibrium state between water and soil solid phase during the simulation time of 2 days. In LAST, the amount of adsorbed masses depend mainly on the substance concentration in water and the soil solid phase. As long as the solute concentration in the water phase is higher than in the solid phase, solutes are adsorbed until an equilibrium concentration between both phases is achieved (cf. Sect. 3.1). This means that sorption is also dependent on factors that can disturb this equilibrium state, to enable further sorption. One such factor could be solute mass loss in the soil solid phase due to degradation, which, if not accounted for as in this special case, leads to a stable equilibrium state once it is achieved. Thus, at the end of the simulations with different parameterizations, the amount of solute masses and their distribution are almost always equal, independent of the $K_f$ value. The magnitude of the $K_f$ value alone only determines how fast the equilibrium state arises. For the simulation with small $K_f$ values, an equilibrium state in all soil depths is reached approximately 1 day after pesticide application, while only about 144 minutes are required for the simulation with the high $K_f$ values.

Simulations with both, retardation and degradation (Fig. 5b), reveal that degradation is dependent on sorption and the $K_f$ value. This is because we assume that degradation only occurs as long as solutes are adsorbed in the solid phase. This shows the general mutual dependence of sorption and degradation processes.

## 6.2 Indicators for solute over-mixing from the 7-day plot-scale simulations

Despite a reasonably good match of simulated and observed bromide mass profiles after 7 days (Fig. 6a), we find indications that an over-mixing of solutes (cf. Sect. 2.2.2) could occur in LAST over longer time scales. While the described "deep shift" and accumulation of bromide masses in soil depths between 30-55 cm (cf. Sect. 5.3.1) could reflect the uncertainty of soil hydraulic properties like the saturated hydraulic conductivity $K_s$, the long mass tail underneath the mass accumulation in the gleyic subsoil around 70 cm depth probably results from artificial over-mixing. Note that the low-permeable gley horizon in this depth has a $K_s$ value of the order of $10^{-8}$ m s$^{-1}$, which implies highly stagnant conditions and thus strongly reduced advective particle movement. Nevertheless, particle diffusion (driven by the random walk, Eq. 5) still occurs due to the particle density and thus water content gradient in this depth originating from the particle accumulation above the gley horizon. Particle diffusion entails diffusive transport of solute masses into deeper soil depths. However, this mass transport might be too strong in our model, as the perfect mixing of solute masses between all water particles in a grid element (Sternagel et al., 2019), leads to small, systematic errors in each time step. These errors accumulate on the 7-day scale and result in over-predictions of the displaced solute masses transported by the diffusing water particles (Green et al., 2002). In particular, the subsequent infiltration of pure water particles with zero solute concentration has the potential to "flush out" solutes, leading to the clear tailing of bromide masses even deeper than 1 m (Fig. 6a). Also, the second simulation with a refined soil domain discretization $dz$ of 0.05 m entails this too strong solute displacement, which shows that over-mixing cannot be simply avoided in our model by using a finer vertical Euler-Grid discretization as sometimes suggested (e.g. Boso et al., 2013). An even finer discretization would lead to huge, excessive simulation times because the finer soil discretization has the consequence that also the time steps become smaller to fulfil the Courant criterion and a much higher number of water particles would be needed. Without a higher particle number, there would be too little particles in the single soil layers to distribute them to the bins properly and to ensure a numerically and statistically valid random walk.

We argue that in natural soils, solutes spread diffusively across water stored in different pore sizes (Kutílek and Nielsen, 1994). Hence, diffusive movement into and out of these pores, as well as their entrapment, depend strongly on the pore size. This implies a time scale for solute mixing among waters in different pore sizes and a flushing-out process that is significantly larger than assumed in our perfectly-mixed approach.

The results of the 7-day and 21-day simulation with exactly the same model setup but with activated reactive transport for IPU does not show any indication for over-mixing (Fig. 6b). This is probably due to the retardation and degradation processes that hinder, or mask, a possible over-mixing as solute masses are adsorbed to the soil solid phase and degraded. Based on the previous findings, we can only assume that the resulting 7-day and 21-day IPU mass profiles are also "deep shifted" due to over-mixing; but comparable data are required for analysis, especially on larger time scales of several weeks.

## 6.3 Comparison of plot-scale simulations over different periods and simulation of FLU breakthrough on the field site

Comparing the plot-scale mass profiles of the 2-day, 7-day and 21-day periods reveals remarkable differences. Regarding bromide transport, the longer drainage phase after irrigation during the 7-day period implies that water and dissolved bromide have more time to redistribute and diffuse through the soil, compared to the 2-day period.

This is reflected in the mass accumulation above the gley horizon observed in the plot-scale experiments at site P4 (cf. Fig 5a). Furthermore, as mentioned in section 4.2.2, IPU can indeed exhibit *DT50* values of just a couple of days in natural soils, which is surely relevant when comparing periods of 2, 7 and 21 days. This is reflected in the higher relative IPU degradation amount of 41.5 % in the 7-day period (cf. Fig. 6b) and 73.3 % in the 21-day period

(cf. Fig 5c) compared to just 13 % in the 2-day period (cf. Fig 4b) for the strong reactive transport parameterization (cf. Table 2). Bending and Rodriguez-Cruz (2007) found in their experiments a remaining mean IPU mass, relative to input amount, of around 65 % in soil samples between 0-80 cm depth after 20 days. The 21-day simulations for IPU (Fig. 6c) result in relative, remaining IPU masses between approximately 30-90 % in the depth 0-80 cm for the strong and weak RT parameterization, respectively. Additionally, Bending et al. (2003) found in further

experiments a remaining mean IPU mass of around 39-55 % of input mass in the first 15 cm of soil at four sites after 21 days. LAST in turn simulates relative, remaining IPU masses between 31-94 % in the depth 0-15 cm for the strong and mean RT parameterization, respectively. Hence, the 21-day simulations of IPU transport produce relative, remaining IPU mass ranges in different soil depths, which seem to be reasonable as they include these observed results.

The results of the FLU breakthrough simulations reveal the general difficulty of simulating the temporal dynamics of the breakthrough curve observed in the tile drain (cf. Fig. 7a and 7b). This corroborates that the use of one single parameter configuration in this study (cf. Table 4) for the simulation of FLU breakthrough on the entire field site, is too simple to capture the obviously much higher spatial heterogeneity of the 400 m² field. This is in line with

experimental findings of Klaus et al. (2014), who reported a strongly variable transport within five distributed soil profiles, which they have examined on this field. Nevertheless, the selected average parameterization of $K_f$ and *DT50* (cf. Sect. 4.2.4, Table 4) allows reasonable simulation of i) adsorption of FLU, especially during the first, rainfall-driven phase when water with FLU infiltrates and redistributes, as well as ii) degradation of FLU, particularly during the period of three weeks between the two irrigation phases. This is reflected in the

remobilization of FLU in the second irrigation phase. As LAST was previously able to calculate adsorption and degradation of FLU in a suitable magnitude, the remobilized, cumulative FLU masses are in turn in the range of observations in the second irrigation phase (cf. Fig. 7d). The steady-state assumption after the first irrigation phase (cf. Sect. 4.2.4) is in line with the observation of Klaus et al. (2014) that the initial background runoff in the tile-drain tube is approximately regained after 500 minutes.

We thus conclude that the reactive transport method implemented in LAST, with the simple parameter configuration, is sufficient to reproduce FLU concentrations and masses that leached into the tile drain and the observed remobilization during the second irrigation phase, three weeks after FLU application. Klaus and Zehe (2011) obtained similar results by using the 2-D model CATFLOW to simulate observed breakthrough of bromide and IPU into a tile-drain at a nearby site in the Weiherbach catchment.

**6.4 General reflections on Lagrangian models for solute transport**

In line with other Lagrangian models using particle tracking for solute transport (e.g. Delay and Bodin, 2001; Berkowitz et al., 2006), our approach shares common assumptions and characteristics. Either particles represent solutes or as in LAST, water parcels, which carry solute masses through the soil domain; and simultaneously, the

particles are not independent but interact with each other as well as with the soil domain by sorption and degradation.

However, LAST has important differences compared to some other particle-based models (e.g. Engdahl et al., 2017; Engdahl et al., 2019; Schmidt et al., 2019), which have been published recently as an alternative to the common solute transport approaches discussed in the introduction. These models calculate mass transfers between particles of different substance species to represent mainly chemical reactions, while our Lagrangian approach calculates the mass transfer of a single substance species among water particles, as well as between water particles and the adsorbing soil phase to represent solute mixing and chemical sorption, respectively. However, by just comparing the implementation of mass transfers in the other models and LAST, regardless of the different application purposes, there is an important difference. These other particle-based models do not use a spatial discretization of the soil domain (Euler-Grid) to determine the spatial proximity or affiliation of particles, and to describe mass transfers between them. Instead, they use a co-location probability approach, which describes solute particle interactions like mass transfer based on a reaction probability dependent on the distance between particle pairs. This approach has advantages in simulating transport and reactions of multiple substances on larger spatial scales of geochemical systems like aquifers, compared to the use of an Euler-Grid. It also offers advantages to overcome the described artificial over-mixing problem of Eulerian control volumes (cf. Sect. 1, 6.2). However, this approach also has drawbacks. For example, one drawback of the miRPT algorithm of Schmidt et al. (2019) is related to its transfer process of solutes. In this process, all eligible solute masses must ultimately be transferred from mobile particles (= water phase) to immobile particles (= soil solid phase) to calculate degradation. Subsequently, the residual, undegraded masses are again transferred back to the water phase for further transport. This implies that masses move between the phases without being subject to degradation or adsorption, which is computationally less efficient because a sufficient spatial distribution and a large number of immobile particles is necessary. In both approaches, miRPT and LAST, solute reactions like degradation are calculated only for the immobile particles. However, due to the use of a spatially discretized soil domain, the reactive solute transport method in LAST is, in contrast, able to perform specific calculations for the partial mass transfer between water and soil solid phase. This is more efficient for transport simulations at the 1-D plot scale, and is less time-consuming and computationally intensive than the approach of the miRPT algorithm. Furthermore, these Lagrangian particle-tracking approaches ultimately require a spatial discretization to calculate solute concentrations, which they achieve by grouping of adjacent particles within a specifically defined radius. This approach is thus similar to soil domain discretization of Eulerian methods, which justifies the Euler-Grid in LAST. In general, the extended LAST-Model with an accounting for reactive solute transport requires only a moderate increase in simulation times compared to the originally published model version (Sternagel et al., 2019). A total simulation time of only 20 to 30 minutes on a moderately powerful PC (Intel Core i7, 3400 MHz, 32 GB RAM) is required for simulations at the heterogeneous site 10 over 2 days, which we consider reasonable relative to the improved model functionality and physical soundness.

**7 Conclusions and Outlook**

Overall, the main findings of this study are that:

- Simulation results demonstrate the feasibility to simulate reactive transport of solutes, through partially saturated soils, within a Lagrangian model framework (cf. Sect. 6.1.1).
- Comparisons to results of HYDRUS 1-D underline that the structural macropore domain is an asset of LAST, which enables an accounting of preferential bypassing and re-infiltration of solutes (cf. Sect. 6.1.2). This is also crucial for predicting preferential leaching of reactive substances under the influence of the effects of sorption and degradation.
- LAST shares common assumptions with other alternative particle-based models but has beneficial characteristics for the simulation of reactive solute transport in partially saturated soil plots (cf. Sect. 6.4).
- 7-day plot-scale simulations show that, while the current formulation yields reasonably good results for bromide transport, some over-mixing of solutes via diffusion is present (cf. Sect. 6.2).
- 21-day plot-scale simulations reveal a reasonable behaviour of reactive IPU transport on larger time scales, also quantitatively compared to results of experiments (cf. Sect. 6.3).
- FLU breakthrough simulations prove the ability of the Lagrangian approach to estimate the remobilization of adsorbed reactive substances on a field site in a second irrigation phase three weeks after application (cf. Sect. 6.3).

Taken together, these findings verify the relevance and innovation of the presented reactive solute transport method in a Lagrangian approach. To the best of our knowledge, no other particle-based Lagrangian framework has applied reactive transport in this way before to simulate sorption and degradation processes at the transport of reactive substances through partially saturated soil plots, even under preferential flow conditions, as well as the breakthrough and remobilization of pesticides on a field site.

In future work, we intend to address possible improvements to the LAST formulation, to better quantify solute transport over longer time scales. One option would be to perform long-term soil column experiments to examine how tracers and pesticides diffusively enter and leave different pore sizes. Based on such experiment results, one could improve the solute transport routine to better account for mixing between water particles that are stored in pores of different size. The Lagrangian approach offers promising opportunities in this regard, as it distinguishes particle movements in different velocity bins, which represent water in different pore sizes (cf. Sect. 2). In this way, it may be possible to simulate, in each time step and grid element, the solute mass exchange between water particles using a specific diffusive transfer rate that is dependent on the pore size or bin in which the particles are stored. With this approach, we would overcome the perfect mixing assumption and may apply pore size-specific sorption with a bin-dependent gradient of $K_f$ values.

*Data availability.* The previously published version of the LAST-Model (Sternagel et al., 2019) is already available in a GitHub repository: https://github.com/KIT-HYD/last-model (Mälicke and Sternagel, 2020). We also intend to provide the extended model version of this study with reference data and the presented test experiments in this repository. Otherwise, please contact Alexander Sternagel (alexander.sternagel@kit.edu).

*Author contributions.* AS wrote the paper, did the main code developments and carried out the analysis. JK and EZ provided the data. RL, JK, BB and EZ contributed to the theoretical framework and helped with interpreting and editing.

*Competing interests.* The authors declare that they have no conflict of interest.

*Financial support.* The article processing charges for this open-access publication were covered by a Research
Centre of the Helmholtz Association.

**Appendix: Detailed description of the macropore domain of LAST**

The following descriptions and the equations in Table A1 should complement the presentation of the macropore
domain of LAST in Sect. 2.2.2 and Fig. 2 as well as serve to understand better the model theory. In this context, it
is important to recall that we have already introduced the macropore domain of LAST in our previous study
(Sternagel et al., 2019).

*Structure of macropore domain*
LAST offers a structured macropore/preferential flow domain (*pfd*) consisting of a certain number of macropores.
Each macropore has the shape and structure of a straight circular cylinder with a predefined length $L_M$ (m) and
diameter *dmac* (m) containing spherically shaped particles (cf. Fig. 2a) (Sternagel et al., 2019). The
parameterization of the preferential flow domain bases on observable field data, such as the mean numbers of
macropores of certain diameters, their hydraulic properties, and length distribution. These structural data can be
directly obtained from field observations or inverse modelling with tracer data, but must not be spatially resolved
because LAST operates on the 1-D scale. From these observable parameters, it is further possible to calculate
additional *pfd* parameters like the total volume, stored water mass at saturation, the circumference $C$ (m) and the
flux rate. The total number of macropores at a study site is classified and distributed over three depth classes (big,
medium or small) to allow for a depth-dependent mass exchange with the matrix domain. To calculate water
contents and tracer concentrations, the macropores of the *pfd* are vertically subdivided into grid elements of certain
length $dz_{pfd}$ (m). Similar to the matrix domain, water contents and solute concentrations are also regarded as
averaged over these grid elements (Sternagel et al., 2019). Within a grid element of a macropore, a certain number
of particles is packed, each having a mass and being geometrically defined by a diameter and volume. These
properties can be derived from the total water mass and predefined number of maximum possible particles stored
in a fully saturated macropore as well as the water density.

*Infiltration and macropore filling*
At the upper boundary of the soil domain prevails a variable flux condition dependent on the incoming precipitation
intensity. First, the incoming precipitation water mass ($m_{rain}$) accumulates in a fictive surface storage from which,
subsequently, infiltrating water masses into the matrix ($m_{matrix}$) and the *pfd* ($m_{pfd}$) and related particle numbers are
calculated (cf. Eqs. A1-A3 in Table A1). The presented equations refer to masses and not fluxes as LAST generally
works with discrete particles and their masses. The actual water content and the flux densities of the topsoil control
infiltration and distribution of water particles to both domains. The two processes are further determined by the
matric potential gradient and hydraulic conductivity of the topsoil matrix (following principles of Darcy's law),
together with the friction and gravity within the macropores. After the infiltration, macropores are filled from the

bottom to the top, comparable to the filling of an empty bottle with water (cf. Fig. 2b), by assuming purely advective flow in the macropore domain as we assume a steady state balance between gravity and dissipative energy loss by friction at the macropore walls. This advective macropore flow is determined by the hydraulic conductivity $k_{pfd}$ (m s$^{-1}$) in a macropore. Zehe and Flühler (2001) measured the velocity of water flow in undisturbed soil samples from the Weiherbach catchment (cf. Sect. 4.1) dominated by macropore influence. They found a clear proportionality of macropore flux rate and the square of macropore radius $\frac{dmac}{2}$ (m), which can be described by a linear relationship. This leads to the calculation of $k_{pfd}$ (m s$^{-1}$) (cf. Eq. A5 in Table A1) under the assumption that the macropore flux rate and hydraulic conductivity as well as the advective velocity of a water particle in a macropore are equal as we presume purely gravity-driven flow.

*Exchange between macropores and matrix*

Interactions at the interface between *pfd* and matrix with the exchange of water particles and thus also solutes are assumed to be mainly driven by matric potential gradients and hydraulic conductivity of both domains, which depend on an exchange length and flow velocities in the respective domains. We assume that exchange is only possible from the saturated parts/grid elements of the *pfd* into the matrix as expecting that the purely advective downward flow of water in macropores is much larger than lateral exchange fluxes.

As described above, the total observed number of macropores *nmac* at a study site is distributed over three depth classes. Hence, the total macropore number is multiplied with a distribution factor *f* for big ($f_{big}$), medium ($f_{med}$) and small ($f_{sml}$) macropores, respectively (cf. Fig. Ac). The saturated grid elements (blue filled squares in Fig. 2c) of the respective three macropore classes are coupled due to their depth order. For instance, the red and black framed grid elements of the three macropore classes are respectively coupled because they are saturated and have the same position in depth order. The coupling thereby enables the simultaneous calculation of diffusive water fluxes $q_{mix}$ (m s$^{-1}$) (cf. Eq. A4 in Table A1) out of the respective grid elements of all three macropore classes.

In the current version, LAST works with a no-flow condition at the lower boundary of the *pfd*. For the lower matrix boundary, however, we actually assume a soil domain of 1.5 m length in total, which is larger than the soil space (0-1 m) we concentrate on in the simulations to avoid boundary effects. That means water particles may freely pass the lower boundary depth of 1 m.

**Table A1.** Relevant LAST-Model equations and related parameters.

| Name | Equation | Parameters |
|---|---|---|
| Eq. A1: Incoming precipitation mass $m_{rain}$ (kg) | $m_{rain} = q_{rain} \cdot \rho_w \cdot \Delta t \cdot A$ | $q_{rain}$ (m s$^{-1}$): precipitation flux density; $\rho_w$ (kg m$^{-3}$): water density; $\Delta t$ (s): simulation time step; $A$ (m²): soil plot area |

**Eq. A2:**
Infiltrating water
mass into matrix
$m_{matrix}$ (kg)

$$m_{matrix} = \left(\frac{k\_m_1 + k_s}{2}\right) \cdot \left(\frac{\psi_1 - \psi_2}{dz} + 1\right) \cdot A \cdot \rho_w \cdot \Delta t$$

$k\_m_1$ (m s$^{-1}$): hydraulic conductivity of first matrix grid element; $k_s$ (m s$^{-1}$): saturated hydraulic conductivity of matrix; $\psi_1 - \psi_2$ (m): matric potential difference between the surface and first matrix grid element; $dz$ (m): matrix grid element length

**Eq. A3:**
Infiltrating water
mass into pfd $m_{pfd}$
(kg)

$$m_{pfd} = k_{pfd} \cdot \pi \cdot \left(\frac{dmac}{2}\right)^2 \cdot \rho_w \cdot \Delta t \cdot nmac$$

$k_{pfd}$ (m s$^{-1}$): hydraulic conductivity of a macropore; $dmac$ (m): diameter of a macropore; $nmac$ (-): total number of macropores within pfd

**Eq. A4: Mixing**
flux between pfd-
matrix $q_{mix}$(m s$^{-1}$)

$$q_{mix} = \frac{2 \cdot k_s \cdot k\_m_i}{(k_s + k\_m_i)} \cdot \frac{\psi_i}{dmac} \cdot C \cdot dz_{pfd}$$

$k_s$ (m s$^{-1}$): saturated hydraulic conductivity of matrix; $k\_m_i$ (m s$^{-1}$): current hydraulic conductivity of the respective matrix grid element; $\psi_i$ (m): matric potential of the actual matrix grid element; $dmac$ (m): macropore diameter; $C$ (m): circumference of a macropore grid element; $dz_{pfd}$ (m): length of macropore grid element

**Eq. A5: Hydraulic**
conductivity pfd
$k_{pfd}$ (m s$^{-1}$)

$$k_{pfd} = 2884.2 \cdot \left(\frac{dmac}{2}\right)^2$$

$\frac{dmac}{2}$: macropore radius

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
