# Peer review of "Simulation of reactive solute transport in the critical zone: A Lagrangian model for transient flow and preferential transport"

_Hydrology and Earth System Sciences, 2020_

## Referee Comment (RC1) · Anonymous Referee #1 · 12 Nov 2020

This manuscript reports on the incremental development of a Lagrangian (random-walk) method originally proposed by Zehe and Jackisch (HESS 2016) to simulate water transfers in the unsaturated zone. This model was then adapted by Sternagel et al. (HESS 2019) to simulate both water and non-reactive mass transfers in double-porosity media (soil matrix and macropores). The developments presented here relate to the additional implementation of non-linear sorption and first-order degradation processes. This extended model is then applied to non-reactive (bromide) and reactive (Isoproturon) field tracer experiment data. Beyond the various comments I make below about i) the theoretical-methodological background and ii) the model application examples, I believe that the implemented developments, i.e. non-linear sorption and

first-order degradation processes, are not substantial enough to warrant a new publication in HESS. I therefore recommend a major revision of the manuscript. Below are my additional comments.

MAIN COMMENTS

**1. Model equations and simulation algorithm. Missing from the manuscript are the equations of the double-porosity flow and (reactive) transport model which are supposed to be solved (simulated) by the proposed Lagrangian method. While it is clear that the original Lagrangian method by Zehe and Jackisch (2016) was developed to solve the Richards equation, the flow and transport equations equations associated with the extended Lagrangian method are not specified either in the article of Sternagel et al. (2019) or in this manuscript. Without these equations, it is difficult to assess the soundness of the LAST-model framework. What I particularly miss is the mathematical description of water and solute exchanges between the preferential flow domain and the soil matrix domain. The modeling equations for reactive transport processes provided in this manuscript are themselves not self-contained. Let us consider the example of equation (6) which describes the sorption reactions. According to this equation, the mass of reactive solute can only decrease over time. Although it is mentioned that this equation only describes the adsorption process, the equation describing desorption and the coupling between the two equations should also be provided so that the term mrs(t) does not only decrease over time. In addition, I think it would be useful to detail the entire Lagrangian algorithm step by step.**

**2. Diffusive transport vs. diffusive mixing. As correctly mentioned in P5L1-2 (page 5, lines 1-2), the Lagrangian algorithm described by equation (5) integrates an advective transport term and a diffusive transport term. But the description given P6L8-11 does not seem to be consistent with this equation. The authors discuss the advective displacement of the particles, followed by a redistribution of mass between the different particles that are in the same Eulerian control volume. This mass redistribution is referred to as "diffusive mixing" by the authors. I have two concerns here. On the one**

[Figure]

hand, the part of the transport described by the second term of equation (5) does not seem to be reflected here, and on the other hand I believe that the expression "diffusive mixing", used at various places in the manuscript, is not appropriate because it could be wrongly confused with diffusive transport. I suggest replacing "diffusive mixing" by "particle mixing" or any other expression that the authors might consider appropriate.

**3. Parameter meaning and values used in the simulations. A number of simulation parameters listed in Tables 1, Table 2, and Table 3 are only very briefly described in the table captions, e.g. alpha, n, and since these parameters do not appear in any of the equations in the manuscript, nor apparently in the equations of the 2009 article, it is difficult to assess the relevance of these parameters. Nor is it specified whether the values indicated in Tables 1-3 correspond to fixed known values, or are empirical (only specified for the Kf and DT50 IPU parameters), or whether these parameters have been estimated through a calibration process to best fit the model against observations. The same question applies for the macropore Ks value (P13L25-26) and for the parameters that control the mass exchanges in the Hydrus dual-domain simulations. It is therefore difficult to assess the comparisons between the LAST and HYDRUS simulations shown in Figure 4c.**

**4. Relevance of the simulations provided to illustrate the model's reactive transport simulation capabilities. The yellow profile in Fig. 3a is barely visible. It may be necessary to indicate in figure caption that this yellow profile (simulation taking only sorption into account) and the light blue profile (simulation taking both sorption and degradation into account) overlap. And given this overlap, the statement P15L10 is incorrect: Figure 3a does not show "significant retardation and degradation", as there is no difference between sorption only and sorption associated with degradation. The similarities between figures 4a (sorption only) and 4b (sorption associated to degradation) also suggest a weak influence of degradation, i.e. adding this process in the simulations does not seem to significantly improve the model fit on the observed data. Therefore, the relevance of the data sets used to illustrate the capacity of the LAST model to simulate**

degradation processes is questionable. I do not question the implementation of degradation in the Lagrangian method, which is actually conventional and straightforward, but the use of this option seems not very relevant with respect to the selected datasets as it does not allow to significantly improve the simulation of real profiles. Similarly, the low sensitivity of the model with respect to the sorption coefficient Kf, as shown in Fig. 4a and acknowledged P21L34-35, also raises questions about the relevance of the data sets used to illustrate the model's reactive transport simulation capabilities. I therefore suggest applying the model to other (more relevant) experimental data to better illustrate the interest of the model add-ons.

**5. Long-term simulations. As stated in the introduction, P3L18-21, one of the main objectives of this work is to assess the ability of the LAST model to perform long-term simulations. It is later clarified that "long term" refers to a period of 7 days, where short term refers to a period of two days (P10L25-27). Beyond the questions that could be raised about whether or not the difference in duration between these periods is significant, I wonder more generally about the capacity of such a model to simulate flow and transport over longer durations involving a modification of the soil structure over time. I think it would be interesting to add a few lines on this topic in the manuscript.**

**6. Over-Mixing. The authors mention possible over-mixing artefacts in their simulations at long times (one week duration), but this hypothesis is described as uncertain, e.g. P20L18-19, P22L18-19, P22L35-36. Yet this type of problem is supposed to be easily identifiable. Simulations should be repeated using a more refined spatial (Eulerian grid) discretization and the results compared. Why has this not been done? I believe it is important to fix this question, and not to relegate it to a future study as suggested in the concluding remarks (P24L6).**

OTHER SPECIFIC COMMENTS

**7. P2L15-17. It should be acknowledged that the laminar flow assumption applies equally to the LAST model.**

[Figure]

**8. Something is missing to understand the transition from equation (3) to equation (5). On the one hand it is expected, according to the classical formalism of RW, to have theta(t) instead of theta_r+i*delta_theta, and on the other hand the term Theta_r is not described in the following lines P5L2-4. One must wait for line 13.**

**9. P5L5-6. According to the way equation (5) is written, the random number Z should not be drawn from a uniform distribution between -1 and 1 but from a standard normal distribution.**

**10. P7L12-13. This sentence suggests that retardation coefficients are only used with Eulerian models whereas their use is also common with Lagrangian methods.**

**11. Much of section 6.3 "General reflections on Lagrangian models for solute transport" would be better placed in the introduction.**

**12. P23L15-18. The fact that with the miRPT method the degradation reactions are restricted to immobile particles is presented as a drawback... but I do not see the difference with the authors' Lagrangian method. If I understand correctly what is written in P6L36, P8L19, and in chapter 3.2, the degradation reactions are also restricted to the adsorbed phase... Please clarify.**

**13. Appendix. I do not think this Appendix is useful. What is reported here corresponds to results already published, i.e. the reader can find the figures A1 and A2 with the related information in the articles of Zehe and Jackisch (2016) and Sternagel et al. (2019). I suggest deleting the Appendix and referring directly to the articles in question.**

MINOR COMMENTS AND TYPO ERRORS

**14. P3L30. The term "actual" does not seem to fit well here, please try to reword it.**

**15. P5L12-13 and Eq. (5). NB should be N (or vice versa)**

**16. Parameter units in equations. Please consider changing the specific units (kg, s, etc.) into generic units (M, T, ...). This will avoid unit conversion factors like 86400 in**

equation 8.

**17. P13L19-20. Remove the quotes from the ref. Gerke and van Genuchten 1993.**

**18. Conservative vs. reactive solute. At various places in the manuscript, the term "conservative" is used as opposed to "reactive", e.g. P16L24-25, P17L26-27. It might be better to replace "conservative" by "non-reactive" because a solute may be prone to sorption reactions (therefore reactive), but not to degradation reactions (therefore mass conservative).**

**19. P17L17. It is not clear here what "particle-bound transport" means. I found the explanation later in the manuscript, P21L26-28, but this content should be moved here.**

**20. P23L29. Unclear what is a "moderately powerful PC".**

**21. P25L6 and Fig. A1. Please consider changing the term "naive" to "classical" or "standard" (more neutral) when referring to the RW method.**

---

## Author Comment (AC1) · 23 Nov 2020

**Response to comments of Anonymous Referee #1**

On behalf of all co-authors, I sincerely thank the Anonymous Referee #1 for the thoughtful and detailed assessment of our work.

*R1: Beyond the various comments I make below about i) the theoretical-methodological background and ii) the model application examples, I believe that the implemented developments, i.e. non-linear sorption and first-order degradation processes, are not substantial enough to warrant a new publication in HESS.*

**AS:** Thank you for this critical comment. We are not aware of any studies that combine depth-dependent, non-linear sorption and first-order degradation with a particle-based approach for reactive solute transport in the unsaturated zone. Thus, we think that the presented developments of LAST are indeed innovative and relevant. However, we agree that from a mathematical point of view a simulation of reactive transport means just to account for two additional processes represented by quite simple equations for sorption and degradation. However, this does not imply that predictions of reactive transport phenomena are only marginally more complex than simulations of conservative transport. The two additional processes of sorption and degradation increase predictive uncertainty, as reflected in the sensitivity of the simulated Isoproturon profiles to variations of reactive transport parameters (cf. Figure 4b in the manuscript). Moreover, the implementation of sorption into the LAST framework is not straightforward as a retardation factor doesn't help in our approach (cf. comment below). The proposed way to account for sorption is hence pretty novel, and we show that this approach allows feasible predictions of reactive pesticide transport under different, transient flow conditions (well-mixed, preferential flow); also compared to HYDRUS. However, we admit that further testing of this approach is desirable and we look for additional experimental data and we will discuss the novelty of our approach in the revised manuscript more clearly to show the significance to the reader.

Main comments

*R1: Model equations and simulation algorithm. Missing from the manuscript are the equations of the double-porosity flow and (reactive) transport model which are supposed to be solved (simulated) by the proposed Lagrangian method. While it is clear that the original Lagrangian method by Zehe and Jackisch (2016) was developed to solve the Richards equation, the flow and transport equations equations associated with the extended Lagrangian method are not specified either in the article of Sternagel et al. (2019) or in this manuscript. Without these equations, it is difficult to assess the soundness of the LAST-model framework. What I particularly miss is the mathematical description of water and solute exchanges between the preferential flow domain and the soil matrix domain. The modeling equations for reactive transport processes provided in this manuscript are themselves not self-contained. Let us consider the example of equation (6) which describes the sorption reactions. According to this equation, the mass of reactive solute can only decrease over time. Although it is mentioned that this equation only describes the adsorption process, the equation describing desorption and the coupling between the two equations should also be provided so that the term mrs(t) does not only decrease over time. In addition, I think it would be useful to detail the entire Lagrangian algorithm step by step.*

**AS:** Thank you for this important comment. Basically, the flow and transport equation of the extended LAST-Model is equal to the Fokker-Planck equation derived by Zehe and Jackisch (2016). The only

difference is that we additionally assign a solute mass to each water particle and hence, the solutes are also advectively and diffusively/dispersively displaced (Eq. 5), together with the water particles (cf. comment below). Thus, we do not assume a second particle species representing solutes but tag the water particles by a solute mass and we do not have implemented a new, specific equation for the transport of solutes. Furthermore, in Sternagel et al. (2019) we show in Eq. 6 how the exchange of water particles, and hence solute masses, between matrix and macropores is calculated. In the current manuscript, we explain these points; but in the revision, we will add further (brief) explanation to ensure clarity. We also provide the complete model code on GitHub, if someone is interested in having a closer look at the Lagrangian algorithm and the numeric. We think that presenting and explaining the entire algorithm in detail within the manuscript would exceed the scope of the study. However, we will complement Eq. 6 of the present study with a further equation for desorption to make the implementation of the sorption process clearer.

*R1: Diffusive transport vs. diffusive mixing. As correctly mentioned in P5L1-2 (page 5, lines 1-2), the Lagrangian algorithm described by equation (5) integrates an advective transport term and a diffusive transport term. But the description given P6L8-11 does not seem to be consistent with this equation. The authors discuss the advective displacement of the particles, followed by a redistribution of mass between the different particles that are in the same Eulerian control volume. This mass redistribution is referred to as "diffusive mixing" by the authors. I have two concerns here. On the one hand, the part of the transport described by the second term of equation (5) does not seem to be reflected here, and on the other hand I believe that the expression "diffusive mixing", used at various places in the manuscript, is not appropriate because it could be wrongly confused with diffusive transport. I suggest replacing "diffusive mixing" by "particle mixing" or any other expression that the authors might consider appropriate.*

**AS:** You have generally recognized our concept of the mass redistribution among water particles. Eq. 5, however, is actually only applied for the displacement of water particles but as the particles carry solute masses, these solutes are also advectively and diffusively/dispersively displaced in each time step, together with the water particles. On page 6, lines 8-11, we no longer refer to Eq. 5, and instead just describe how solute masses are redistributed among the water particles after the advective-dispersive displacement of Eq. 5. We will use a different expression for this "diffusive mixing" of solutes among water particles to avoid confusion.

*R1: Parameter meaning and values used in the simulations. A number of simulation parameters listed in Tables 1, Table 2, and Table 3 are only very briefly described in the table captions, e.g. alpha, n, and since these parameters do not appear in any of the equations in the manuscript, nor apparently in the equations of the 2009 article, it is difficult to assess the relevance of these parameters. Nor is it specified whether the values indicated in Tables 1-3 correspond to fixed known values, or are empirical (only specified for the Kf and DT50 IPU parameters), or whether these parameters have been estimated through a calibration process to best fit the model against observations. The same question applies for the macropore Ks value (P13L25-26) and for the parameters that control the mass exchanges in the Hydrus dual-domain simulations. It is therefore difficult to assess the comparisons between the LAST and HYDRUS simulations shown in Figure 4c.*

**AS:** The soil hydraulic parameters (like alpha, n etc.) are the van Genuchten-Mualem parameters, which define the soil hydraulic properties of the soil and can be used to calculate the matric potential of soil at a certain water content, for example. As the van Genuchten-Mualem concept and its parameters are generally well known and established in unsaturated soil physics/hydrology, we do not

describe them further. However, we will add a respective citation, when introducing these parameters. Additionally, the residual parameters in Table 1 and 3 are observed, measured data from the described experiments. In Sternagel et al. (2019), we provide a detailed description of how the observed data are processed for use in our model (e.g. macropore data) and how sensitive our model is to the uncertainty range of observed data (e.g. to the saturated hydraulic conductivity $Ks$). However, we will modify the text to improve clarity in the respective passages.

*R1: Relevance of the simulations provided to illustrate the model's reactive transport simulation capabilities. The yellow profile in Fig. 3a is barely visible. It may be necessary to indicate in figure caption that this yellow profile (simulation taking only sorption into account) and the light blue profile (simulation taking both sorption and degradation into account) overlap. And given this overlap, the statement P15L10 is incorrect: Figure 3a does not show "significant retardation and degradation", as there is no difference between sorption only and sorption associated with degradation. The similarities between figures 4a (sorption only) and 4b (sorption associated to degradation) also suggest a weak influence of degradation, i.e. adding this process in the simulations does not seem to significantly improve the model fit on the observed data. Therefore, the relevance of the data sets used to illustrate the capacity of the LAST model to simulate degradation processes is questionable. I do not question the implementation of degradation in the Lagrangian method, which is actually conventional and straightforward, but the use of this option seems not very relevant with respect to the selected datasets as it does not allow to significantly improve the simulation of real profiles. Similarly, the low sensitivity of the model with respect to the sorption coefficient Kf, as shown in Fig. 4a and acknowledged P21L34-35, also raises questions about the relevance of the data sets used to illustrate the model's reactive transport simulation capabilities. I therefore suggest applying the model to other (more relevant) experimental data to better illustrate the interest of the model add-ons.*

**AS:** Yes, you are mentioning an important point. However, we think that the current experimental database is suitable to particularly show a noticeable effect of sorption, while the time scale of 2 days may be indeed a little too short to explore the role of degradation in its entirety. We will examine another dataset of a breakthrough experiment and check the possibility to adapt the model setup to simulate the solute concentration time series of the breakthrough.

However, the question remains of how to define a significant difference between results. We think that there are indeed at least noticeable differences between, especially, the results of the conservative, reference simulation runs and the simulations performed with full reactive transport, particularly with respect to the spatially small and temporarily short scale.

Regarding the results of the well-mixed simulations presented in Figure 3, we refer here especially to the difference between the conservative, reference simulation and the simulations performed with full reactive transport, which is clearly visible (RMSE difference of 7.3 %). Of course, the influence of degradation is relatively small but still detectable due to the quite high *DT50* value of 23 days and the short time scale. We will rephrase this passage to make it clearer.

Regarding the results of the preferential flow dominated simulations presented in Figure 4, we think that, in particular, the blue highlighted area in Figure 4b indeed shows clear differences in the final mass profiles, when varying the *DT50* value in observed ranges. Thus, the mass profile simulated with a high *DT50* value exhibits the strongest difference to the mass profiles simulated only with retardation in Figure 4a (with a total mass loss of around 0.131 g IPU for an input of 1 g in just two days, which we think is significant).

To summarize, we of course admit that further simulations with long-term datasets would be desirable but the availability of suitable data is scarce. However, we also think that the presented results provide scientifically significant insights and demonstrate the feasibility of the implementation of depth-dependent sorption and degradation into a Lagrangian framework. To the best of our knowledge, there are no other studies implementing depth-dependent, non-linear sorption and first-order degradation into a particle-based Lagrangian approach to simulate reactive solute transport in the unsaturated soil zone.

*R1: Long-term simulations. As stated in the introduction, P3L18-21, one of the main objectives of this work is to assess the ability of the LAST model to perform long-term simulations. It is later clarified that "long term" refers to a period of 7 days, where short term refers to a period of two days (P10L25-27). Beyond the questions that could be raised about whether or not the difference in duration between these periods is significant, I wonder more generally about the capacity of such a model to simulate flow and transport over longer durations involving a modification of the soil structure over time. I think it would be interesting to add a few lines on this topic in the manuscript.*

**AS:** We agree that the expression "long-term" might be misleading and we will change the expression in the revised manuscript. Yet, we think that there are indeed remarkable differences between a simulation period of 2 and 7 days. In particular, the drainage phase after irrigation is longer, which implies that water and dissolved bromide have more time to redistribute and diffuse through the soil. This is reflected in the accumulation above the gley horizon observed in the experiments. Further, as we have discussed in the manuscript, the reactive substance IPU can indeed exhibit DT50 values of just a couple of days in natural soils, which is surely relevant when comparing periods of 2 and 7 days (cf. Fig. 4b and 5b of the manuscript). We will add these clarifications to the revised manuscript to justify the difference between the 2 and 7 days simulations.

However, the soil structure may of course change on larger time scales. In our application cases, this might have no significant influence as we work on quite small spatial scales of around 1 m³ on artificially shaped, agricultural fields. The soils on these fields may have a more persistent structure than natural soils because conditions are better controllable. Up to now, our LAST-Model has not been accounting for a changing soil structure but we think, this could be straightforward to implement, e.g. by just changing the soil hydraulic parameters and macropore data after certain phases in long-term simulations.

*R1: Over-Mixing. The authors mention possible over-mixing artefacts in their simulations at long times (one week duration), but this hypothesis is described as uncertain, e.g. P20L18-19, P22L18-19, P22L35-36. Yet this type of problem is supposed to be easily identifiable. Simulations should be repeated using a more refined spatial (Eulerian grid) discretization and the results compared. Why has this not been done? I believe it is important to fix this question, and not to relegate it to a future study as suggested in the concluding remarks (P24L6).*

**AS:** Thanks for this interesting point. We repeated the long-term simulation using a finer soil domain discretization *dz* of 0.05 m. The results show that the solute over-mixing is indeed slightly mitigated but a too strong displacement below 1 m is still clearly visible (Fig. 1), which we attribute to over-mixing. However, an even finer discretization would lead to huge, excessive simulation times because the finer soil discretization has the consequence that also the time steps become smaller to fulfil the Courant criterion and a much higher amount of particles would be needed. Without a higher particle amount, in the single soil layers would be too little particles to distribute them to the bins properly and

to ensure a numerically and statistically valid random walk. We will add a respective passage to this issue in the revised manuscript and show the simulations results for a finer discretization for comparison.

[Figure]

**Figure 1.** Bromide (Br) mass profiles of 7-days simulation using different soil discretizations *dz* and compared to observed mass profile.

Other specific comments

*R1: P2L15-17. It should be acknowledged that the laminar flow assumption applies equally to the LAST model.*

**AS:** In general, you are right for flow in the matrix domain because the theoretical starting point of the displacement equation (Eq. 5) of LAST is the Richards equation (Eq. 1), as we describe in section 2. Further, we use an adaptive time stepping to fulfil the Courant criterion and to ensure that the particles do not travel further than the length of a grid element *dz* in a time step. We will discuss this issue in the revised manuscript.

*R1: Something is missing to understand the transition from equation (3) to equation (5). On the one hand it is expected, according to the classical formalism of RW, to have theta(t) instead of theta_r+i*delta_theta, and on the other hand the term Theta_r is not described in the following lines P5L2-4. One must wait for line 13.*

**AS:** Yes, you are generally right. With our expression $\theta_r + i \cdot \Delta\theta$ in Eq. 5, we mean the soil moisture of the current bin in a grid element and time step. This is formally equivalent to $\theta(t)$ but a speciality of our binning-approach, which is then described in the subsequent passage (page 5). We will revise and already introduce this formal equivalence of expressions as well as $\theta_r$ directly after Eq. 5.

*R1: P5L5-6. According to the way equation (5) is written, the random number Z should not be drawn from a uniform distribution between -1 and 1 but from a standard normal distribution.*

**AS:** Thank you. You are of course right; it should be a random number from a standard normal distribution. We will revise this.

**R1:** *P7L12-13. This sentence suggests that retardation coefficients are only used with Eulerian models whereas their use is also common with Lagrangian methods.*

**AS:** This sentence specifically refers to our Lagrangian approach. In LAST, there are not two separated particle species for water and solute but we only assume one species of particles/parcels, which represent water and can carry a solute mass. Thus, the travel distance of water particles and solutes cannot be uncoupled, which means that the use of a retardation coefficient is not meaningful in our approach. However, you are right by stating that indeed other Lagrangian approaches use such a retardation coefficient because they may employ two distinct particle species for water and solutes, which is commonly applied in modelling water flow and solute transport in groundwater. We will revise and clarify this passage.

**R1:** *Much of section 6.3 "General reflections on Lagrangian models for solute transport" would be better placed in the introduction.*

**AS:** Of course, some general aspects of this section could also be mentioned in the introduction but we think that they better fit into the discussion, as the reader knows all the details of LAST after reading the entire manuscript and can better compare it to the other new particle-based models, which are described in section 6.3.

**R1:** *P23L15-18. The fact that with the miRPT method the degradation reactions are restricted to immobile particles is presented as a drawback... but I do not see the difference with the authors' Lagrangian method. If I understand correctly what is written in P6L36, P8L19, and in chapter 3.2, the degradation reactions are also restricted to the adsorbed phase... Please clarify.*

**AS:** Sorry for the misunderstanding. The drawback is not related to the restriction of degradation to immobile particles but to the transfer process of solute masses as described on page 23, lines 15 ff.: *"However, this approach also has drawbacks. For example, the miRPT algorithm of Schmidt et al. (2019) transfers all solute masses from mobile particles (= water phase) to immobile particles (= soil solid phase) for reaction and subsequently back to the mobile particles for further transport."*

What we mean here is the fact that in the miRPT model all solute masses must be ultimately transferred from the water phase to the solid phase to calculate degradation in each time step. Subsequently, the residual, not degraded masses are again transferred back to the water phase. This approach is quite consuming because most masses are moved there and back between the phases without being subject to degradation or adsorption. In LAST, we use specific calculations for the transfer of masses from water particles to adsorbing phase, whereby just a part of solute masses is transferred (= adsorbed). Only these adsorbed masses are then subject to degradation and the residual, not degraded masses stay in the adsorbing phase until the concentration gradient between water and adsorbing phase may turn and desorption occurs. Further, you are of course right by arguing that LAST also calculates degradation only in the adsorbing phase. We will revise this passage to make this point clearer.

*R1: Appendix. I do not think this Appendix is useful. What is reported here corresponds to results already published, i.e. the reader can find the figures A1 and A2 with the related information in the articles of Zehe and Jackisch (2016) and Sternagel et al. (2019). I suggest deleting the Appendix and referring directly to the articles in question.*

**AS:** Yes, this could be one option but we think that the Appendix might be useful for some readers to conveniently access and compare the main findings of the previous LAST studies.

Minor comments

**AS:** Thank you for the further minor comments. Most of them seem clear and constructive. We will check and consider changing the manuscript, accordingly.

*R1: P13L19-20. Remove the quotes from the ref. Gerke and van Genuchten 1993.*

**AS:** Please note that this is not a reference but the actual name of the selected dual-permeability approach in HYDRUS.

*R1: P25L6 and Fig. A1. Please consider changing the term "naive" to "classical" or "standard" (more neutral) when referring to the RW method.*

**AS:** Here, we just use the same formulation as Zehe and Jackisch (2016) to be consistent.

Thank you very much,

Alexander Sternagel on behalf of all authors

**References**

Zehe, E., and Jackisch, C.: A Lagrangian model for soil water dynamics during rainfall-driven conditions, Hydrology And Earth System Sciences, 20, 3511-3526, 10.5194/hess-20-3511-2016, 2016.

Sternagel, A., Loritz, R., Wilcke, W., and Zehe, E.: Simulating preferential soil water flow and tracer transport using the Lagrangian Soil Water and Solute Transport Model, Hydrology And Earth System Sciences, 23, 4249-4267, 10.5194/hess-23-4249-2019, 2019.

---

## Referee Comment (RC2) · Anonymous Referee #2 · 30 Nov 2020

Comments on the paper HESS-2020-527 entitled "Simulation of reactive solute transport in the critical zone. A Lagrangian model for transient flow and preferential transport ", by A. Sternagel et al.

This contribution proposes an interesting Lagrangian technique employing random walkers to solve both unsaturated flow and solute transport in a one-dimensional soil column. My understanding is that the so-called LAST model has been already published in HESS, but the present work would improve the model by giving it a preferential transport module in the form of advection in a macropore compartment. It also faces model results to actual experiments of water and reactive solute percolations from the

surface in various soil columns.

The key fact that would justify the eventual publication of the paper is in the Lagrangian approach to flow (and incidentally transport), which is not very much widespread in the ongoing literature. The Richards equation is manipulated to form a non-linear Fokker-Planck-Kolmogorov equation (FPKE), then solved by classical advection-dispersion conveyed random walkers. These represent water parcels, to which a solute mass is added with the aim of solving transport. As there is no other set of particles specifically allotted to solve transport, the transport process is mainly advective but accounts for solute dispersion-diffusion by a "full mixing" process which states that all particles mapped onto a discrete cell of the soil column will share after a given time step the same solute mass, as the mean of the mass beard by all the particles within the cell. . . The process is slightly rough, but, provided that the time spent in the cell is enough, and provided that the macropore compartment is not assigned the same diffusive process, the whole is receivable.

The major concerns I have with the present writing are mainly associated with lacks in the depiction of the model. I remind that this model is the key justification for providing an innovative study. The authors often refer to the previous work by Sternabel et al. 2019, but the present writing becomes sometimes fuzzy and not self-containing, with the meaning that any reader should understand at first glance how physically the model works without relying upon multiple other readings.

The model manipulates particles as water parcels that, I guess, bear an elementary mass (or a volume) of water. Because of the non-linearity of the Richards equation, the particles have to be periodically mapped onto a regular grid discretizing the modeled domain, the aim being to evaluate the local water content, which in turn conditions the motion of the particles. It is unclear how the particles are mapped. If it is only their number, then this number needs for calibration for being transformed into a water content. If the particles are mapped as mass or volume, then compared to the local open pore space, then a water content is affordable.
It is clear that for small one-dimensional systems, the boundary conditions deeply influence both flow and transport within the system. Nothing is told about these boundary conditions, e.g., prescribed infiltration at the top, free drainage at the bottom, how they are handled with particles etc... This does not help to be confident in the reliability of the LAST model.

The implementation of the macropore compartment is very evasive. My understanding is that in wide open pores, the capillary pressure is almost zero, which makes that the open void is either water-saturated of empty. This feature changes the physics of flow and could also change that of transport, for example, with a "mixing" process that does not apply in the macropores. It is not clear at all how the macrospore compartment is calculated. It is stated that the macropore compartment is only filled up with "event particles" (rapid infiltration?) I do not see why. It is also stated that there exist some exchanges between particles in the matrix and in the macropore compartments. How does that work? Is there any probability for an elementary particle to fall, for a given cell, within either the matrix of the macropore? What is the transition probability for a particle to pass from one compartment to the other, etc. The physics handled and how it translates into algorithms should be detailed, all the more this macropore compartment seems to be an improvement of the LAST model compared with previous attempts.

The present writing is always referring to Sternagel 2019, to the point that, for example, parameters reported in tables 1 and 3 are not documented. It is clear, for a non-novice reader, that these parameters inherit from the Van-Genuchten model linking effective conductivity and capillary pressure with water content. This is not a reason to not mention the Van-Genuchten model, the significance of parameters, and with what they are associated. Otherwise, the reading may become cumbersome.

By the way, I also find the Appendices useless, since results have already been published and do not really fit the aim of the present writing.

Finally, the application of the LAST model to actual transport data is interesting, but
needs moderate rewriting to be picky on what the model does and what it does not. Otherwise, it is easy to fit model outputs onto data, but without saying too much on the way the model outputs are acquired. For the rest, nothing really pivotal to mention; the authors do their best for mimicking experiments, which is always a hard task.

Other suggestions.

P.2, lines 8-12. I do not understand why dual porosity approaches would not work, but macropores within a matrix would work. In both cases, this is a representation merging two compartments with interactions between both.

P.3, line 18. Mention the characteristic times of what is stated as being a long-term simulation

P.4, line 24. State that z is counted positive upward.

P.4, Eq. (4). Rigorously speaking, it should be – v (and not +v) which corresponds to K/theta – dD/dz. Otherwise, Eq (3) is not a FPKE.

P.4, Eq. (5). In the second term (within brackets) of the RHS of (5), the sign in front of dD/dz should be a minus sign.

P4, Eq. (5) notation Deltai not explained at this stage.

P.5, line 5. z should be a number drawn from a Gaussian distribution of zero mean and unit variance. It is right that a uniform distribution of z can render Gaussian spatial distribution of particles, but only after, say, 20-30 jumps. I would stick to a Gaussian distribution of z, even if in the effective algorithm a uniform distribution can be employed.

P. 6 lines 23 and 27. Matric? Is it meant matrix?

P.8, Eq. (6). Slightly rearranging eq. (6) results in a concentration on the LHS compensated in the RHS by a concentration to the power beta. The Freundlich coefficient cannot be dimensionless. By the way, I do not see why a retardation factor associated with Euter approaches to sorption could not be applied to Lagrangian models. Incidentally, Eq (6) is very similar to a kinetically-controlled Freundlich sorption process. Has the time step dt in LAST any influence on the sorption. If instantaneous equilibrium is suggested (as mentioned by discussing on a retardation factor), then (6) has no meaning because the mobile concentration in a given cell should equilibrate with that onto the solid phase... I fear that I do not understand.

P.8, Line 7. Which form is given to the concentration onto the solide phase? Is that the so-called reactive mass in (6)?

P. 9, Eq. (8, 9) Do not use k as a notation in (8) and (9), simply because the constant has not the same significance as in (7). Or refer to k in (7) as a value in h-1 and specify that (8) is a first-order approximation of the exponential in (7) for small arguments of the exponential.

P.12. Line 21, Specify how much water volume (mass) corresponds to the initial number of 2 million particles.

P. 14, 15, tables 1 and 3... Van Genuchten parameters? Specify it both in the text and in the table captions.

P. 18, lines 1-5. As suggested above, it seems that the Freundlich adsorption is mainly insensitive to the Freundlich coefficient in an equation close to a x-order kinetics... Too large time step for moving the particles from one cell to the other? Too rapid kinetics, thus rendering instantaneous equilibrium?. If so, why not to discuss simply on local equilibrium?

Next Pages... With data, hardly available, a thing that I understand, it is hard to take all the wording in Sections 5 and 6, as not partly conjectured... Fitting an accurate model onto sparse data will never inform on the model reliability, its capability of mimicking actual systems, and is sensitivity to parameters. This point does not jeopardize publication; it simply underlines that LAST needs for confrontations with synthetic test cases... Perhaps a concluding remark to add, of an appendix to rapidly build

---

## Author Comment (AC2) · 8 Dec 2020

**Response to comments of Anonymous Referee #2**

On behalf of all co-authors, I sincerely thank the Anonymous Referee #2 for the thoughtful and detailed assessment of our work.

Comments

*R2: My understanding is that the so-called LAST model has been already published in HESS, but the present work would improve the model by giving it a preferential transport module in the form of advection in a macropore compartment. […] The major concerns I have with the present writing are mainly associated with lacks in the depiction of the model. I remind that this model is the key justification for providing an innovative study. The authors often refer to the previous work by Sternabel et al. 2019, but the present writing becomes sometimes fuzzy and not self-containing, with the meaning that any reader should understand at first glance how physically the model works without relying upon multiple other readings.*

**AS:** We appreciate that you find the study interesting and thank you for pointing out that it presents a substantial innovation of the particle-based Lagrangian approach. As detailed below, we will provide the requested necessary details, particularly to better explain the functioning of the macropore domain and its mass exchange with the soil matrix. However, we will do this without duplicating the entire model description as the macropore domain was already introduced in our previous study (Sternagel et al., 2019). The present manuscript provides an expanded validation of the LAST-Model and its macropore domain by exploring the influence of preferential flow, together with the effects of sorption and degradation on reactive solute transport. To further stress the capability of the framework to cope with reactive transport, we will evaluate the possibility to include additional test simulations.

*R2: The model manipulates particles as water parcels that, I guess, bear an elementary mass (or a volume) of water. Because of the non-linearity of the Richards equation, the particles have to be periodically mapped onto a regular grid discretizing the modeled domain, the aim being to evaluate the local water content, which in turn conditions the motion of the particles. It is unclear how the particles are mapped. If it is only their number, then this number needs for calibration for being transformed into a water content. If the particles are mapped as mass or volume, then compared to the local open pore space, then a water content is affordable.*

**AS:** We apologize for being unclear about this in the current manuscript version and we will improve this in the revised manuscript. Indeed, we count the number of water particles in each grid element and time step and compute a particle density per soil volume. By multiplying this with the particle mass and water density, we obtain the soil water content. The soil water content is needed to update the drift and diffusion terms in the Langevin equation (Eq. 5). Please note that Zehe and Jackisch (2016) already examined the sensitivity of the approach to different particle numbers. The mass of the pre-event water stored in soil is much larger than the mass of infiltrating event water. To suitably represent this event water relative to the much larger pre-event water fraction and thus, to ensure a sufficient stochastic treatment of particles, the total particle number must be high. Due to the relatively small 1-D extent of our model, the use of 2 million particles is sufficient here.

*R2: It is clear that for small one-dimensional systems, the boundary conditions deeply influence both flow and transport within the system. Nothing is told about these boundary conditions, e.g., prescribed infiltration at the top, free drainage at the bottom, how they are handled with particles etc… This does not help to be confident in the reliability of the LAST model.*

**AS:** We agree that this is of key importance. We will add necessary details to make the treatment of upper and lower domain boundaries transparent in the revised manuscript. At the upper boundary, we apply a variable flux condition depending on rainfall input. The incoming water mass is transferred into a corresponding number of water particles and these are distributed between the macropore and matrix domain. The infiltration capacity of the topsoil matrix is determined by its water content, the corresponding hydraulic conductivity and the driving potential gradient. The infiltration capacity of the macropore system is determined by the gravity potential gradient and their very high hydraulic conductivity ($\sim 10^{-2}$ m/s) (cf. Eqs. 3 and 4 in Sternagel et al. (2019)). At the lower boundary, we actually assume a soil domain of 1.5 – 2 m length in total, which is larger than the soil space (0 - 1 m) we actually concentrate on in the simulations to avoid boundary effects. Particles may hence freely pass the depth of 1 m.

*R2: It is not clear at all how the macrospore compartment is calculated. It is stated that the macropore compartment is only filled up with "event particles" (rapid infiltration?) I do not see why. It is also stated that there exist some exchanges between particles in the matrix and in the macropore compartments. How does that work? Is there any probability for an elementary particle to fall, for a given cell, within either the matrix of the macropore? What is the transition probability for a particle to pass from one compartment to the other, etc. The physics handled and how it translates into algorithms should be detailed, all the more this macropore compartment seems to be an improvement of the LAST model compared with previous attempts.*

**AS:** The macropore domain has been introduced and extensively tested in our previous study (Sternagel et al., 2019). However, we of course agree that the manuscript needs to provide more details here, as the structural macropore domain is a main part of LAST and a clear difference to other approaches like HYDRUS. In the macropores, we assume purely advective flow that is driven by gravity only, as capillary forces are neglectable. We indeed assume that advection in macropores is much faster than exchange with the soil matrix and that macropores gradually fill from their bottom to the top. Exchange of water particles between saturated parts of the macropores with the surrounding matrix is calculated by means of the Darcy law (cf. Eq. 6 in Sternagel et al. (2019)) based on hydraulic conductivity gradient and the matric potential of the matrix. In each time step, the calculated exchange flux is based on its water mass converted into a corresponding number of particles, which are transferred from the macropore to the matrix domain in the respective depths. The entire approach and its sensitivity was tested in Sternagel et al. (2019).

*R2: Finally, the application of the LAST model to actual transport data is interesting, but needs moderate rewriting to be picky on what the model does and what it does not. Otherwise, it is easy to fit model outputs onto data, but without saying too much on the way the model outputs are acquired. For the rest, nothing really pivotal to mention; the authors do their best for mimicking experiments, which is always a hard task.*

**AS:** Thank you for this important comment. Of course, LAST has various adjustable parameters and hence bears several degrees of freedom but actually, we did not fit the model parameters to match the observed results of our shown test cases. The used parameters in Tables 1 – 3 are all observed data

from the experiments or empirical data from the PPDB database or other studies. We will try to provide clarification in the revised manuscript, but we also think that an extensive repetition of the model structure and parameterization from our previous study would be redundant and out of the scope of this study. We further think that referencing to mainly one other study is reasonable for a reader to obtain more detailed information about the model. In this regard, please see also our very last response/comment.

Other suggestions

*R2: P.2, lines 8-12. I do not understand why dual porosity approaches would not work, but macropores within a matrix would work. In both cases, this is a representation merging two compartments with interactions between both.*

**AS:** Generally, you are of course right. The difference is that in classical dual-domain approaches there are just theoretical descriptions of a macropore domain by applying a second, higher permeability for a part of matrix flow or assuming most of water is stagnant compared to a small flowing fraction mimicking macropore flow. However, in LAST we use a physically based macropore domain, which contains geometrically defined macropores with a certain amount, diameter, length etc.. The geometrical properties are important as they determine the macropore volume and thus scale exchange fluxes between macropores and matrix. We will provide more information on the model setup and especially on the macropore domain and its interactions with the matrix.

*R2: P.3, line 18. Mention the characteristic times of what is stated as being a long-term simulation*

**AS:** Thanks. Actually, we do this at the beginning of the methods section but in the revised manuscript, we will avoid using the expression "long-term" and just state the actual duration in days when referring to the different simulations.

*R2: P.4, line 24. State that z is counted positive upward.*

**AS:** Thank you. We will add this information.

*R2: P.4, Eq. (4). Rigorously speaking, it should be – v (and not +v) which corresponds to K/theta – dD/dz. Otherwise, Eq (3) is not a FPKE.*

**AS:** We agree and will revise the manuscript accordingly.

*R2: P.4, Eq. (5). In the second term (within brackets) of the RHS of (5), the sign in front of dD/dz should be a minus sign.*

**AS:** The term $\frac{\partial D(\theta_r + i \cdot \Delta \theta)}{\partial z}$ in Eq.5 corrects the downward drift term in the case of a spatially variable diffusion and hence, is added with a plus-sign as upward velocity, contrary to the advection/drift term, in line with Roth and Hammel (1997). We will clarify this in the revised manuscript.

*R2: P4, Eq. (5) notation Deltai not explained at this stage.*

**AS:** Yes, you are right. The first reviewer also noted that. We will revise the passage after Eq. 5 and add explanation.

*R2: P.5, line 5. z should be a number drawn from a Gaussian distribution of zero mean and unit variance. It is right that a uniform distribution of z can render Gaussian spatial distribution of particles, but only after, say, 20-30 jumps. I would stick to a Gaussian distribution of z, even if in the effective algorithm a uniform distribution can be employed.*

**AS:** Yes, you are right. Again, also the first reviewer noted that mistake. We indeed use random numbers drawn from a normal distribution but just misstated that in the manuscript. We will revise that.

*R2: P. 6 lines 23 and 27. Matric? Is it meant matrix?*

**AS:** To the best of our knowledge, matric potential and matrix potential are synonyms.

*R2: P.8, Eq. (6). Slightly rearranging eq. (6) results in a concentration on the LHS compensated in the RHS by a concentration to the power beta. The Freundlich coefficient cannot be dimensionless. By the way, I do not see why a retardation factor associated with Euter approaches to sorption could not be applied to Lagrangian models. Incidentally, Eq (6) is very similar to a kinetically-controlled Freundlich sorption process. Has the time step dt in LAST any influence on the sorption. If instantaneous equilibrium is suggested (as mentioned by discussing on a retardation factor), then (6) has no meaning because the mobile concentration in a given cell should equilibrate with that onto the solid phase… I fear that I do not understand.*

**AS:** In general, you are right. *KF* and *beta* are both empirical constants that determine the shape and slope of the sorption isotherm of a certain substance. They are both often described as dimensionless coefficients but *KF* can actually adopt different units/forms to balance the units of the equation, particularly when *beta* is not equal to 1. I think a dimensionless representation of *KF* is quite common, but in our case, it actually must have the unit $\left(\frac{kg}{kg}\right)^{\frac{1}{beta}}$. We will add this information.

Further, regarding the retardation factor we here specifically refer to our Lagrangian approach. In LAST, there are not two separated particle species for water and solute but we only assume one species of particles/parcels, which represent water and can carry a solute mass. Thus, the travel distance of water particles and solutes cannot be uncoupled, which means that the use of a retardation coefficient is not meaningful in our approach. However, you are right by stating that indeed other Lagrangian approaches use such a retardation coefficient because they may employ two distinct particle species for water and solutes, which is commonly applied in modelling water flow and solute transport in groundwater. We will revise and clarify this passage.

Moreover, the sorption process indeed indirectly depends on the time step as solute concentrations of water particles vary between time steps and thus also the solute amount, which is adsorbed from a particle in a time step. That means we do not assume an instantaneous equilibrium between water and solid phase in the sorption process. The equilibrium state is achieved after a certain time scale depending on the selected KF parameter, as discussed in the discussion on page 22, line 4 ff..

*R2: P.8, Line 7. Which form is given to the concentration onto the solide phase? Is that the so-called reactive mass in (6)?*

**AS:** The concentration in the adsorbing solid phase is defined as adsorbed mass in kg per m³ of soil. We will provide another equation for the desorption process to clarify this issue.

*R2: P. 9, Eq. (8, 9) Do not use k as a notation in (8) and (9), simply because the constant has not the same significance as in (7). Or refer to k in (7) as a value in h-1 and specify that (8) is a first-order approximation of the exponential in (7) for small arguments of the exponential.*

**AS:** Thank you. We will use a different notation in Eq. 8 and 9.

*R2: P.12. Line 21, Specify how much water volume (mass) corresponds to the initial number of 2 million particles.*

**AS:** This depends on the initial water content of the soil domain. The initial water content is converted to a total water mass (with soil domain volume and water density) and this mass is then divided by the total number of particles (here: 2 million). In this way, the particles in the matrix are defined by a certain water mass. We will clarify that in the revised manuscript.

*R2: P. 14, 15, tables 1 and 3… Van Genuchten parameters? Specify it both in the text and in the table captions.*

**AS:** Thank you, you are right. Sorry for not referencing the van Genuchten-Mualem parameters. We will add a more detailed description of these parameters.

*R2: P. 18, lines 1-5. As suggested above, it seems that the Freundlich adsorption is mainly insensitive to the Freundlich coefficient in an equation close to a x-order kinetics… Too large time step for moving the particles from one cell to the other? Too rapid kinetics, thus rendering instantaneous equilibrium?. If so, why not to discuss simply on local equilibrium?*

**AS:** Here, we mean that the magnitude of the $KF$ value alone only determines how fast the equilibrium state between water and solid phase is achieved (not instantaneous, as stated above). Only without degradation, the adsorbed masses for different $KF$ values are almost equal once this equilibrium state is achieved and maintained because of missing degradation, and thus independent/insensitive to $KF$ in this special case.

*R2: Next Pages… With data, hardly available, a thing that I understand, it is hard to take all the wording in Sections 5 and 6, as not partly conjectured… Fitting an accurate model onto sparse data will never inform on the model reliability, its capability of mimicking actual systems, and is sensitivity to parameters. This point does not jeopardize publication; it simply underlines that LAST needs for confrontations with synthetic test cases… Perhaps a concluding remark to add, of an appendix to rapidly build*

**AS:** Thank you for this good point. Indeed, it is hard to find and apply experimental data to properly test a model and as we also stated in the response to Reviewer #1, we do not really have further reliable datasets, which would not need any fitting. However, the parameterization of the shown test cases and results were not excessively fitted as the described irrigation experiments provided detailed data, which we need for our model (e.g. soil hydraulic properties, initial conditions, irrigation data, macropore data). Of course, the parameterization of LAST bears several degrees of freedom to fit model results to observed data (as any other model), but we only did this in a suitably assumed uncertainty range of observed parameters (e.g. the saturated hydraulic conductivity, as discussed in Sternagel et al. (2019)). Therefore, we think that the presentation and discussion of results is not conjectured but we will check sections 5 and 6 and revise passages, which could convey this impression. Further, we will check for further suitable tests to underline the feasibility of the extended LAST-Model.

Thank you very much,

Alexander Sternagel on behalf of all authors

**References**

Roth, K. and Hammel, K.: Transport of conservative chemical through an unsaturated two-dimensional miller-similar medium with steady state flow, Water Resour. Res., 32, 1653–1663, 1996.

Sternagel, A., Loritz, R., Wilcke, W., and Zehe, E.: Simulating preferential soil water flow and tracer transport using the Lagrangian Soil Water and Solute Transport Model, Hydrology And Earth System Sciences, 23, 4249-4267, 10.5194/hess-23-4249-2019, 2019.

Zehe, E., and Jackisch, C.: A Lagrangian model for soil water dynamics during rainfall-driven conditions, Hydrology And Earth System Sciences, 20, 3511-3526, 10.5194/hess-20-3511-2016, 2016.

---

## Author Response (AR2)

**Dear Editor,**

we sincerely thank you for your professional and constructive coordination of the review process. We are also grateful for the comments of the reviewer, which helped to improve our manuscript. In line with our responses to the reviewer, we thoroughly revised our manuscript to address your and the reviewer's recommendations as outlined below:

- We changed the title of section 4 to better indicate that we test our model mainly with experimental data.
- We revised the passage in section 6.4 to clarify the comparison of the implementation of mass transfers in LAST and the other particle-based Lagrangian models.

Attached to this author's response you find:

- a marked-up manuscript version (all changes or new contents are highlighted in yellow)
- the point-by-point response to both reviewers as published in HESS discussion.

Thank you very much.

Best regards

Alexander Sternagel on behalf of all authors

**Marked-up manuscript version**

[revised manuscript text omitted]

**Point-by-point response to reviews**

**Response to comments of Anonymous Referee #3**

On behalf of all co-authors, I sincerely thank the Anonymous Referee #3 for the thoughtful and detailed assessment of our work.

**Comments**

*R3: Section 4 is termed "Model Benchmarking". I find this confusing on several accounts. Firstly, the "benchmarking" results are reported in Section 5 (Results) while Section 4 reports on simulation setup, and experimental data. Secondly, and more importantly, the comparison of the numerical model results with experimental data is not a benchmarking of the model. I would expect that the model be benchmarked against and/or (i) analytical solutions (if available in simplified setup), (ii) numerical results from other well established codes (simulating exactly the same scenario), (iii) fully controlled analogue experiments (for controllably the same scenario as the numerical model scenario). The provided benchmarking does not meet any of these criteria, but applies the (unbenchmarked) model for the interpretation of concentration data from field experiments in heterogeneous soils where other processes than the ones represented in the numerical model may also be important. Thus, I would recommend benchmarking the numerical model against known solutions (analytical or numerical), and rename Sections 4 and 5 with a title that reflects that the model is used for the interpretation of experimental data.*

**AS:** We thank the reviewer for pointing this out. We admit that the title of section 4 is a little confusing, as we do both in our study. We (i) compare LAST with experimental data and (ii) benchmark it using the well-established HYDRUS 1-D model to simulate the same irrigation experiments with IPU mass profiles at sites 5 and 10. Of course, we would prefer to benchmark the model against analytical solutions describing preferential flow and reactive transport through partially saturated soils. Unfortunately, there is neither an analytical solution nor a commonly accepted numerical model available for this. We admit that there have been applications of Eulerian models to simulate reactive transport of, e.g. pesticides or tracers into tile drains, as discussed in one of our previous studies (Klaus and Zehe, 2011). However, our effort to simulate the breakthrough of IPU with the CATFLOW model, using a 2-D structured, heterogeneous domain, performed worse (Klaus and Zehe, 2011). We will change the title of section 4 to better indicate that we mainly test our model with experimental data.

*R3: Section 6.4: "Instead, they use a co-location probability approach, which describes solute particle interactions like mass transfer based on a reaction probability dependent on the distance between a pair of particles." This passage refers to the reactive random walk particle tracking approaches of Benson and Meerschaert and others. The feature mentioned by the authors is relevant for chemical reactions between different mobile chemical species. The approach described by the authors only considers reactions for a single mobile species (sorption/desorption, first-order degradation). Thus, it is not clear what the authors want to say with this discussion. Do the authors refer to the implementation of diffusive mass transfer between fluid particles or chemical reactions?*

AS: Thank you. You are right by stating that the mentioned approaches by Engdahl et al., 2017; Engdahl et al., 2019; Schmidt et al., 2019 calculate mass transfers between particles of different solute species to represent chemical reactions. Instead, our approach calculates the mass transfer of one solute species between water particles and the adsorbing soil phase to represent sorption and degradation. However, in this passage we especially refer to (i) the process of transferring masses itself, regardless of the purpose of this mass transfer (chemical reactions or sorption/degradation); and (ii) to the general method of how they determine the spatial proximity/affiliation of particles using this co-location probability approach and not an Euler-Grid. We addressed this in the revised manuscript (page 28, line 5 ff.).

Thank you very much,

Alexander Sternagel on behalf of all authors